# Recent Development of Tunable Optical Devices Based on Liquid

**DOI:** 10.3390/molecules27228025

**Published:** 2022-11-18

**Authors:** Qi Wu, Hongxia Zhang, Dagong Jia, Tiegen Liu

**Affiliations:** 1School of Precision Instrument and Opto-Electronics Engineering, Tianjin University, Tianjin 300072, China; 2Key Laboratory of Optoelectronic Information Technical Science, Tianjin University, Tianjin 300072, China

**Keywords:** tunable, liquid, liquid crystal, optical device, beam steering, adaptive liquid lens, optical filter, design

## Abstract

Liquid opens up a new stage of device tunability and gradually replaced solid-state devices and mechanical tuning. It optimizes the control method and improves the dynamic range of many optical devices, exhibiting several attractive features, such as rapid prototyping, miniaturization, easy integration and low power consumption. The advantage makes optical devices widely used in imaging, optical control, telecommunications, autopilot and lab-on-a-chip. Here, we review the tunable liquid devices, including isotropic liquid and anisotropic liquid crystal devices. Due to the unique characteristics of the two types of liquids, the tuning principles and tuning methods are distinguished and demonstrated in detail firstly and then some recent progress in this field, covering the adaptive lens, beam controller, beam filter, bending waveguide, iris, resonator and display devices. Finally, the limitations and future perspectives of the current liquid devices are discussed.

## 1. Introduction

With the devices’ tunability becoming more and more convenient for our lives, it has gradually become an indispensable parameter in optical devices. Due to the limitations of solid materials, the traditional tuning method mainly focuses on the mechanical, and the mechanical contact leads to a short life, low tuning efficiency and slow tuning speed of device, which also limits the miniaturization of optical devices. At the same time, the fixed structure and performance of a solid device mean that they have a high dependence on wavelengths, and the research expansion of wavelengths in the optical field, which is mainly caused by nonlinear optical materials [1] and the continued progress on material structures [2], is a disastrous challenge to solid optics. In recent years, researchers have introduced liquid into optical research. Based on its unique advantages [3], such as flexible adjustability, good compatibility, small size, easy manufacturing, etc., liquid devices can flexibly adjust optical characteristics so that the performance of liquid devices, especially the tunability, is favored by more and more researchers.

Currently, the core liquid mainly includes anisotropic liquid crystals and isotropic ordinary liquids. Liquid crystal materials are widely used, especially in the field of biochemistry. In 2022, Sulayman A. et al. [4] reviewed the applications of liquid crystals as biosensors for molecules [5], protein or peptide detection [6,7,8,9] and disease biomarkers [10]. The liquid crystal materials can also be found in the fields of optical [11,12,13], new energy [14] and the physical [15,16]. The wide application of liquid crystal materials is closely related to the continuous development of liquid crystal molecules. Liquid crystal materials have grown from nothing, from nematic liquid crystals to cholesteric liquid crystals doped with azobenzene chiral molecules [12,13,17,18,19], to the discovery of various derivative liquid crystal materials such as PDLC, SPLC and LCE, with the liquid crystal solubility and helical twisting ability improving [11,20,21]. Based on the liquid crystal molecules, the tuning performance of electrically controlled liquid crystal devices can be guaranteed gradually. There are review articles that explain the types and development of some typical liquid crystal devices and liquid devices before then. In 2018, Yi-Hsin Lin [22] systematically introduced the classification and comparison of adaptive liquid crystal lenses. Shin-Tson Wu et al. [23] summarized the principle of the LC beam steering device and the research progress over the past 19 years. Wang Lei [24] briefly described the research status of liquid crystal materials and devices in the field of terahertz and discussed the development trend of the combination of liquid crystal technology and terahertz technology. Zepeng Zhou [19] gave a comprehensive review of flexible LCP technologies covering electric circuits, antennas, integration and packaging technologies, front-end modules, MEMS and biomedical devices. Dissimilar to LC, due to the isotropic characteristics of ordinary liquid, tunable controls are more focused on the interface control of immiscible liquids. In recent years, devices that use the external environment to control the refractive index profile of liquids have gradually become a research hotspot, mainly including electric control and temperature control. The commonly used liquid includes ethanol and glycol because of their appropriate relative permittivity and stable properties. There are also good review articles in the research of ordinary liquid devices, including the summary of in-plane optical fluid tunable liquid lens in 2018 [25]. To make it easy to sense, the principle has been figured out (see Figure 1a), and the common and unique types optical devices are also summarized (see Figure 1b). Specific details about the principle will be illustrated in next section.

The characteristics and control methods of the two liquids are different, but the ultimate aim is the same: to achieve miniaturization and wide tuning of optical devices. Therefore, this paper briefly summarizes and supplements the development of liquid devices in the past 5 years. In this review, the advantages of liquid devices and the inevitability of their future development are introduced firstly; secondly, the relevant classifications and alignment technologies of liquid crystal materials and the control methods of ordinary liquids are illustrated; then, we introduce the tunable optical devices of liquid crystals in the past five years (2018–2022), including adaptive optical lenses, optical filters and absorbers, optical switching, beam steering, liquid crystal lasers and bistable state devices, with the structures or performance described briefly; at the same time, the optical devices made of ordinary liquid are introduced simply according to the same way; finally, the challenges and prospects of liquid optical devices are discussed.

## 2. Principles of Liquid Optics Control

The liquids currently used to make tunable devices include liquid crystals, which can be used to make lenses, filters, wave plates and other devices that can be tuned and transformed by refractive index transformation. Compatible or incompatible liquids that can be used to make lenses and optical switches, optical prisms, curved waveguides, etc. are tunable devices controlled by the liquid–liquid interface and the refractive index gradient distribution of the mixed liquid. The working principle and basis of various liquid optical devices will be introduced below.

### 2.1. Liquid Crystal Control

#### 2.1.1. Liquid Crystal Material

The discovery of liquid crystals can be traced back to 1888, when the turbid state of benzoic acid cholelithol was observed by the Austrian botanist Reinitzer and the German physicist Lehmann successively [26]. Liquid crystals were officially confirmed, and the research on them began. According to the order of molecular arrangement to distinguish the liquid crystal phase, it can be divided into three categories: nematic, cholesteric and sematic and disk. Here, we mainly introduce the nematic and cholesteric liquid crystals that are widely used [27,28,29].

The characteristic of nematic liquid crystal is that the molecules do not have fixed positions, but the orientation of the molecules basically takes the same direction—that is, the long-range orientation is ordered. In the use of liquid crystals, polyimide film is often coated on the surface of the substrate as the initial alignment layer of the liquid crystal material. When no driving voltage is applied, the alignment of the directors of the liquid crystal molecules tends toward a predetermined direction, so that the initial free energy of the liquid crystal molecules is minimized. After a driving voltage of a certain frequency is applied to the electrodes, due to the anisotropy of the dielectric constant and conductivity of the liquid crystal, the liquid crystal molecules are subjected to a force that changes the orientation of the molecular axis.

Within the elastic limit of the liquid crystal, this electric field causes, I torque rotates the molecular axis, and finally, the surface anchoring force of the liquid crystal is balanced with the action of the external electric field. The liquid crystal is regarded as a kind of modulator of the optical system—that is, when the current (power) is input into the liquid crystal, the effective refractive index of the liquid crystal molecules will change, and as a result, the light beam passing through the liquid crystal layer is modulated. Due to the anisotropic properties of liquid crystal molecules, the relationship between the effective refractive index and the liquid crystal deflection angle θ can be expressed as [27]
(1)$$n_{eff}(y)=\frac{n_{o}n_{e}}{\sqrt{n_{o}^{2}\cos\phi (y)+n_{e}^{2}\cos\phi (y)}}.$$

Cholesteric liquid crystals are also called helical liquid crystals. Since the difference between the cholesteric liquid crystal and the nematic liquid crystal is that the alignment of the molecules changes direction in a helical manner along one axisIthe distance between two adjacent points that are aligned on the axis with a difference of 2π is called the piI the schematic diagram of the arrangement of helical liquid crystal molecules. The cholesteric liquid crystal has a spontaneous helical arrangement and can selectively reflect circularly polarized light of the same chirality, forming a forbidden photon band. The central wavelength *λ_c_* of the reflection spectrum of the circularly polarized light selectively reflected by the cholesteric liquid crystal is related to the helical pitch *p* of the cholesteric liquid crystal and the average refractive index of the liquid crystal ($\bar{n}$ = (*n_o_* + *n_e_*)/2), where [30]
(2)$$\lambda_{c}=\bar{n}p\;.$$

The helical structure of the cholesteric liquid product determines its own unique optical properties, including optical rotation, polarized dichroism, Bragg reflection, the production of reflective liquid crystal displays, color display devices, picture polarizers, broadband reflective polarizers, circular polarizing filters, etc. The helical pitch of cholesteric liquid crystals is easily changed by external fields, such as chiral dopant concentration, temperature control, UV curing and applied electric field. Therefore, cholesteric liquid crystals can be used to develop a variety of tunable optical devices. In recent years, cholesteric liquid crystal-based tunable filters, tunable gratings and tunable lasers have become the research hotspots of liquid crystal devices.

#### 2.1.2. Liquid Crystal Alignment Technology

The coating layer in the liquid crystal cell that can induce the molecules from disorderly arrangement to orderly alignment is called the alignment layer, and the liquid cement used to prepare the alignment layer is the alignment agent. At present, polyimide (PI) is mainly used as the alignment layer. After rubbing alignment or photoalignment, PI molecules show orientation anisotropy on the surface, inducing liquid crystal molecules to be “anchored” and directionally aligned under the electric field [31,32]. The rubbing alignment is to conduct unidirectional mechanical friction on the surface of PI layer so as to form dense lines on the surface of the PI layer or induce the alignment of the PI molecular chain. It has the advantages of a simple process, stable LC orientation and mature development of the LC orientation control. However, in recent years, the rubbing alignment technology has been prone to generate dust, static electricity and mechanical damage, which will seriously affect the performance and quality of devices. In view of the defects of the rubbing alignment, many research institutions domestic and abroad have carried out the research on noncontact alignment technologies, such as ion beam alignment, oblique vapor deposition of silicon oxide and photoalignment, having emerged as the times require [32,33]. Chiou et al. [34] formed a periodic dense striated structure on the PI alignment layer through soft embossing technology instead of the traditional rubbing method, and the width limit reached 3 μm.

Among them, photoalignment technology has attracted more attention for its high-resolution surface and curved alignment [32,35]. The photoalignment is to induce the physicochemical reaction of the alignment agents through LPUV light, which leads the directional alignment, and then transfer this order to the LC through intermolecular interactions to achieve orientation control. According to the photochemical reaction mechanism, the photoalignment technology can be divided into three categories: photoisomerization, photodimerization and photodegradation [36,37]. The three technologies all depend on specific structures, such as photoisomerization corresponding to azobenzene structures [38]. The core team research on photoalignment technology is Lu Yanqing and Hu Wei’s group from Nanjing University. The applications of photoalignment in display and photonics are summarized in [39], mainly focusing on geometric phase manipulation:micro-polarizer array using photoalignment technology for image sensors;electrically tunable liquid crystal q-plates;electrically switchable liquid crystal Fresnel lens and gratings;liquid crystal optical elements with integrated Pancharatnam–Berry phases.

In a word, LC is gradually emerging in telecommunications, optical fiber communication systems, sensors, switchable lenses, LC optical converters and other optical fields, not just displays.

### 2.2. Conventional Liquid Devices Control

#### 2.2.1. Interface Control

When the two-phase fluid is in contact with the cell wall, the contact angle of the interface is formed at this three-phase contact point. The angle is related to the surface tension of the three-phase material, and the angle of the meniscus of the contact surface can be changed by some special methods so as to complete the regulation of the surface curvature. These special ways include the electrowetting effect (EW) [40], thermal effect, dielectrophoresis (DEP) [41], etc. The decrease of the surface tension coefficient of a liquid with an increasing temperature is the core principle of thermal effect tuning. What needs to be described in detail are the EW (Figure 2a,b) and DEP(Figure 2c), as both are electrical controls.

In EW lenses, the wetting effect of a hydrophobic surface can be changed by applying a voltage, making the surface more hydrophilic. Therefore, the external voltage applied can alter the contact angle between the liquid and the cell’s sidewall, so as to obtain the droplet curvature and the focal length. Real-time modulation of the focal length can be achieved by changing the applied voltage. DEP is a phenomenon that an object with a low dielectric constant is subjected to force in a nonuniform electric field. DEP drives the liquid with a high dielectric constant to the strong electric field originally occupied by a medium with a smaller dielectric constant. The DEP has nothing to do with whether the object is charged or not but with the size, electrical properties of the object and the surrounding medium, the field strength and its change rate and frequency. By comparison, the core distinguishes between EW and DEP, which is the response liquid, and that EW corresponds to the conductive liquid while DEP corresponds to the dielectric.

#### 2.2.2. Refractive Index Profile Due to Diffusion

For the optical device based on the interface control, the refractive index distribution of the liquid is uniform, and there is no gradient distribution. However, based on the refractive index gradient distribution, many optical devices with better characteristics can be produced. For example, the Luneburg lens has better light focusing and aberration, and the gradient fiber can self-focus to increase the transmission distance. The first GRIN lens based on diffusion was demonstrated in [42] (see Figure 3a,b), which provides a new strategy for developing liquid microlens.

In order to realize the refractive index gradient distribution in liquid, there are two methods currently used: concentration convection diffusion [43] and thermal diffusion [44]. The expression for the diffusion process is [43,44]
(3)$$\frac{\partial C}{\partial t}=D\nabla^{2}C-u\nabla C+S\;,$$
(4)$$\frac{\partial T}{\partial t}=K\nabla^{2}T-v\nabla T\;.$$

The above equations characterize the diffusion at the concentration (*C*) and temperature (*T*), respectively. *D* and *K* represent the diffusion coefficient of the concentration diffusivity and thermal diffusivity of the liquid, respectively. Both *u* and *v* represent the flow rate of the liquid. The *S* parameter represents the contribution of the chemical reaction in the diffusion process. Generally speaking, there is no chemical reaction between the two liquids selected, and it is a passive structure—that is, *S* = 0. The corresponding physical model of the diffusion expression is that the left side of the expression represents the change of concentration and temperature with time. As time goes by, the model will reach a stable state, and the change of concentration and temperature is equal to zero (∂C/∂t=0,∂T/∂t=0).

The first item on the right side represents the gradient diffusion. The concentration and temperature distribution of the substance are not uniform at the beginning, and this gradient distribution forms a diffusion phenomenon; the second item on the right side represents the convection item. The convection term depends on the role of velocity during liquid transport. The two modes of mass transport work together to affect the changes in the substance concentration and temperature. Which mode dominates depends on the parameter Peclet number (*Pe*) [45]. Generally speaking, in order to achieve a better concentration/temperature gradient distribution, the *Pe* number will be relatively smaller.

## 3. Tunable Devices Based on Liquid Crystal

### 3.1. Tunable Lens

#### 3.1.1. Model Liquid Crystal Lens

In order to achieve a relatively uniform refractive index distribution of the liquid crystal tunable lens, a high resistance layer or a dielectric layer will be placed on the electrode, and both contribute to generate a linear change in the voltage and smooth the liquid crystal lens, transforming the phase plane from square to parabolic. Nowadays, modal technology is relatively mature, and lenses with several millimeters in apertures (up to 5–10 mm) and superior uniformity and low voltage control can be produced.

The research of the modal lens has gradually transformed from a single layer to multilayers. The bilayer lens was originally intended to solve the polarization-independent problem, and the specific realization light path diagram was fully elaborated in [46]. In 2019, in order to overcome the dilemma between the tunable range of lens power and the aperture size, Yu-Jen Wang et al. [47] proposed a NLC spatially expanded phase distribution to realize electrically tunable GRIN LC lenses. The lens consists of two LC elements, as shown in Figure 4a. The optical tuning at 0–10 mm is controlled by the upper LC element, and the optical tuning at 10–20 mm is controlled by the lower LC element, and finally achieves 20 mm effective aperture modulation. The individual and combined refractive index profiles are shown in Figure 4b–d. In addition, each LC element can be processed independently, and the focal length of the GRIN LC lens is continuously adjustable. It can be used as a positive lens, a negative lens and a bifocal lens and even has aberration correction function in different spatial positions. In 2020, Kumar, Kang et al. [48] proposed a liquid crystal lens with a three-layer birefringence structure, a layer of uniformly arranged nematic liquid crystal between two PET substrates. The polarization independence is finally achieved by the decomposition of the light beam by the PET layer and the modulation of the extraordinary light by the liquid crystal layer. Subsequent researchers found that the double-layer structure can achieve large aperture and wide tuning of the lens.

In 2022, Li S. et al. [49] fabricated an electrically tunable LALC lens with double-hole electrodes (see Figure 5). The surfaces of the top and middle substrates are engraved with hole electrodes with a diameter of 4 mm and 5 mm, respectively. An ITO common electrode is deposited on the inner surface of the bottle substrate. The lens is set with two layers of electrodes: TELC lenses have a focal length range of 325–220 mm in 24–66 V (DM1); MELC lenses have a focal length range of 310–200 mm in 24–59 V (DM2). The proposed LALC lens can achieve electrical tuning of the focal length by changing the operating voltage, while the two-layer electrode control expands the tuning range of the lens to 325–200 mm.

The array is the core of device integration, and the research of the LC lens array is equally important. In recent years, the Qionghua Wang group has been the main one to study the high-performance LC lens array based on dielectric layers. From 2019 to 2020, three types of double-layer LC lens arrays with composite dielectric layers have been successively proposed. The first two structures are similar but different, as shown in Figure 6a [27] and Figure 6b [50]. Figure 6a shows that the dielectric layer with a low dielectric constant acts as the resistive layer, and the periodically arranged ITOs act as the electrodes. In the voltage-on state, the vertical electric field in the LC layer varies linearly with the lens aperture, and finally, a centrosymmetric refractive index curve is generated. The simulation results show that, by changing the applied voltage, the focal length is tuned from 0.998 mm to infinity. Different from Figure 6a, Figure 6b has two electrodes at the edge of the lens acting as power electrodes. In the middle of the lens, the ITO electrodes above the low dielectric layer and the upper layer of the liquid crystal cell are grounded. The center ground electrode can stabilize the orientation of the liquid crystal above it, and the tilt angle distribution of the liquid crystal molecules can be adjusted by adjusting its width. The third structure is shown in Figure 6c [51], which is a LC lens array with a composited dielectric layer. In design, a spatially nonuniform electric field is generated between the strip electrodes, resulting in a gradient refractive index distribution in the LC layer. Since the upper and lower parts of the LC lens array both adopt a composite dielectric layer, the operation voltage of the LC lens array is effectively reduced. In terms of the LC lens, the double-layer design doubles the phase difference between the center and the periphery of the LC layer, thereby reducing the focal length of the LC lens array. In addition, the shortest focal length (~1.78 mm) of the LC lens array is obtained at V = 3.3 V, and the LC lens array has a large focusing range.

The three LC lens arrays proposed by the Qionghua Wang team have the same advantages: relatively low working voltage, flat substrate surface, simple electrode and uniform LC cavity gap.

A common technique to realize the gradient electric field profile that is required in liquid crystal tunable lenses is the use of a weakly conductive layer or the layer introduced as above. Thanks to this layer, an applied voltage with a certain frequency allows us to obtain a refractive index profile that is required for the lens operation. Due to the limited degrees of freedom, however, it is not possible to avoid aberrations in a weakly conductive layer-based tunable lens for a continuously tunable focal length. Tom Vanackere [52] et al. applied the higher frequency voltage components on the liquid crystal lens and optimized the amplitude and frequency selection of the high-frequency signal through the Zernike polynomial, making the edge refractive index closer to the parabolic distribution, to reduce the lens aberrations drastically.

Due to the birefringence characteristics of liquid crystals, the LC device has a certain dependence on the polarization of the incident light, which hinders the tuning of the LC lens and reduces the light efficiency. In order to improve it, polarization-independent or polarizer-free LC phase modulations are developed, including a double-layered structure, residual phase type, a mixed type (i.e., combination of a double-layered structure and residual phase type) and the type of Kerr effect. However, the two-layers structure must be identical, which is not easy in mass production, and there is also a trade-off between it and the response time. In 2021, Yi-Hsin Lin et al. [53] proposed a novel LC mode for a polarization-independent LC phase modulation by means of orthogonally anisotropic orientations of nematic LC on adjacent microdomains in a single LC layer through the scratched method. Each subdomain is 4 μm, and the total size is 112 μm (Figure 7a). Such an orthogonal (Figure 7b) nematic LC mode includes a subdomain with anisotropic orientations but collectively presents a capability of a polarizer-free optical phase modulation. An OLC mode cell provides a tunable optical phase of 3.35π radians for unpolarized light and different linearly polarized light. A polarizer-free LC micro-lens is also presented.

#### 3.1.2. Pattern Electrode

Although the modal technology is mature, it is difficult to obtain a uniform high resistivity layer as the lens diameter increases. With the development of lithography technology and the improvement of etching precision, complex ITO (indium tin oxide) electrode structures appear. Designing different electrode topologies to realize the refractive index distribution of liquid crystal lenses has gradually become a research hotspot in recent years.

In 2018, Jeroen Beeckman et al. [54] used photolithography to fabricate a discrete multielectrode lens with an electrode width of 12.4 μm and a spacing of 2 μm, and verified the feasibility of the pattern electrode through simulation and experiments. Marenori Kawamura et al. [55] verified the possibility of a tunable conical lens through a similar multi-ring electrodes lens structure; the diameter of the central circular electrode is 183 μm, the width of the ring electrode is 245 μm and the electrode gap is 244 μm. A three-dimensional numerical calculation method for estimating the optical phase retardation in an LC lens is also proposed, and the three-dimensional inclination and azimuth distributions of the theoretically simulated six-ring electrode lens are simulated, the conical phase retardation in the lens area is estimated and the optical phase distribution is experimentally simulated. The results are in good agreement with the numerical results.

Additionally, based on the multi-ring electrode structure, in 2020, J.F. Algorri et al. [56] designed a voltage-divided multi-ring electrode lens, and the width of the electrode and the gap are both 10 μm. This microstructure is based on two main elements, a transmission line acting as a voltage divider and concentric electrodes distributing the voltage across the active area of the liquid crystal evenly (Figure 8a). The concentric electrodes only make electrical contact with the thinner transmission line (R1), not R2, and the main purpose is to control the voltage distribution between V1 and VC. Compared to the modal techniques, the transmission line has a low resistivity. Different geometries of the transmission line will result in different voltage distributions of the electrode at different radii, forming the lenses with different functions. The uniform transmission line (Figure 8b) can form a tapered lens with a linear distribution of a refractive index, and the tapered transmission line (Figure 8c) can form a GRIN lens with a parabolic refractive index distribution, even a logarithmic lens. The magnitude of the voltages V1 and V2 at both ends can also determine the concave or convex lens. In this paper, it is verified that a 1 cm aperture GRIN liquid crystal lens can achieve a focal length variation range of 5.8 D through experiments.

Using the same structure and idea of a transmission line voltage divider, J.F. Algorri et al. [57,58,59] modified the concentric ring electrode into strips with periodic distribution, as shown in Figure 9a. By applying four different voltages: V1, V2, V3 and V4 [60,61,62] to the endpoint of the upper and lower layers, six optical devices were realized. Voltages across the same layer had opposite phases. To avoid the cancellation when the amplitudes are equal between the upper and lower layers, a relative phase shift of 90° was applied. When the electrodes of the bottom substrate are grounded (V1 = V2 = 0), Powell lens and cylindrical lens can be realized according to the shape of the transmission line, such as in Figure 8b,c. If four endpoints work at the same time, the refractive index profile will be axisymmetric. The above two lenses convert to axicons and quasi-parabolic lenses, and the effect of the gap between perpendicular electrodes should be discussed, which relate and influence the function of the beam steering and 2D tunable diffraction grating. To achieve steering towards either the vertical or horizontal direction, a single voltage has to be applied at one terminal, whereas the rest have to be grounded. Combined with specific device parameters, the maximum steering range of the device is from −0.08° to +0.08°. In 2022, a novel pattern electrode [63] is designed to produce a parabolic profile of the voltage difference (Figure 9b). It consists of the generating unit (solid lines) and the distributing unit (dashed lines). The lengths of the resistive lines in the generating unit increase linearly from the center to the edges. The distributing unit is composed of many parallel ITO lines. Detailed voltage distribution derivation means that the terminal voltage can be set according to the required lens parameters. The experiment demonstrated that the optical power changes from −11.1 to +11.3 m^−1^. Some other electrode structures that form parabolic voltage profiles are also demonstrated.

In 2022, Justin Stevens introduced a newly designed electrically tunable gradient index LC lens and its working principle. Based on linear serpentine electrodes [64], no semiconductor layer is required, the width and spacing of the serpentine electrodes are w = 5 µm and g = 5 µm, the two substrates are separated by 40 µm-diameter microspheres and the interlayer is filled with a nematic LC (NLC) mixture with optical birefringence Δ*n* ≈ 0.217. The electric field profile is generated by applying a sinusoidal voltage to each of the terminals: A, B, C and D with the same frequency, and these signals are sequentially phase-shifted by 90°, forming a symmetrical potential profile to control the focal length. The simulation results show that the voltage changes linearly from the edge of the lens to the cente, which preliminarily verifies the feasibility of a lens with 2 mm aperture and realizes an optical power of 9.5 D and a RMS wavefront error of 0.16 µm. And there is something to worth noting, a low-frequency (20 Hz) sine “bias” signal is added to the electrodes to raise the center voltage above the LC threshold, which converts cone voltage to paraboloid voltage. The lens performance was experimentally demonstrated with a miniature camera.

By designing spiral electrodes with different geometrics and densities, an uneven electric field can also be generated to obtain an ideal refractive index profile and realize a tunable focal length. Mingce Chen et al. [65,66] proposed a novel liquid crystal microlens with a planar nonuniform helical microcoil electrode. By applying a driving voltage signal, cylindrical microbeam focusing and spherical microbeam focusing corresponding to circular and elliptical microcoils are achieved, respectively. The entire liquid crystal microlens includes 2 × 2 spiral electrodes, and a pin A with a diameter of 1 mm is located in the center of the spiral electrode; the electrode pin B is connected to four spiral electrodes through a bus. The bus widths in the circular and elliptical electrodes are 30 μm and 20 μm, respectively. The focusing characteristics were experimentally demonstrated. The focal lengths of the circular and elliptical electrodes were 1.75 mm at 50.1 Vrms and 2.35 mm at 55.0 Vrms, respectively.

### 3.2. Tunable Filter and Absorber

Filters span many fields of electromagnetism, including optics and electricity. LC filters are good broadband, anisotropy and electrooptically and magnetooptically modulated. The development of LC filters covers from shortwave (such as visible light, near-infrared band) to longwave (such as mid-to-far infrared, THz band). Therefore, there are many studies on LC filters. The following will classify and introduce them according to wavelength.

#### 3.2.1. Visible Light to Infrared

Although both microwave and infrared belong to electromagnetism, their models are different because of the great difference in wavelengths. The microwave filter is more focused on the improvement of the microstrip line structure, and the early model of the near-infrared filter is the FP cavity. The traditional FP cavity mainly relies on the external mechanical tuning of mems, which usually has the disadvantages of a long response time, complex structure and large volume. The introduction of liquid crystal into FP cavity filter tuning not only improves the response speed but also lays the foundation for the development of device miniaturization.

Since the introduction of liquid crystals, for a long time, the study of a single FP cavity was relatively mature. Jiuning Lin et al. [67] presented an electrically tunable infrared (IR) filter based on a key cascaded liquid–crystal Fabry–Perot (C-LC-FP), working in the wavelength range of 3–5 μm. The C-LC-FP is constructed by closely stacking two FP microcavities with different depths of 12 and 15 μm and fully filled by nematic LC materials. Through the continuous wavelength selection of both microcavities, radiation with a high transmittance and a narrow bandwidth can pass through the filter. According to the electrically controlled birefringence characteristics of nematic LC molecules, the transmission spectrum can be shifted through applying a dual-voltage signal over the C-LC-FP. Compared with common LC-FPs with a single microcavity, the C-LC-FP demonstrates better transmittance peak morphology and spectral selection performance. The experimental results also demonstrated that the number and the shifted scope of the IR transmission peak could be decreased and widened, respectively. Due to the birefringence of liquid crystals, infrared LC-FPs filters are polarization-sensitive, the same as microwave ones, and the polarization of light will greatly affect the filtering characteristics. For the polarization sensitivity of LC-FPs, Zhonglun Liu et al. [68] designed and experimentally verified an infrared filter with a multidirectional layer of PI film, which can make the LC distribute along two mutually perpendicular directions. The filter’s polarization insensitivity is substantially improved.

Besides inheriting the FP cavity structure, the idea of solid optical filters can also be applied to LC filters. Marwan Abuleil et al. [69] combined the Lyot filter structure with liquid crystal, and the Lyot filter transmittance is related to the wavelength. It contains a series of birefringent crystal plates and polarizers. The birefringence axis of each crystal is rotated 45° relative to the axis of the polarizer, and the structure is shown in Figure 10a. The filter tunes between nine spectral bands covering the range 450–1000 nm with bandwidths <10 nm and throughput >80%. Mi-Yun Jeong et al. [70] reported a continuously tunable and bandwidth variable optical notch and bandpass filters created by combining four left- and right-handed circular cholesteric liquid crystal cells without extra optical components. The filter performance was greatly improved by introducing an antireflection layer on the filter device. The filter comprised cholesteric liquid crystal wedge cells with continuous pitch gradients (see Figure 10b). The band wavelength position was spatially tuned from 470–1000 nm. The notch filters are polarization-independent in the spectral ranges. The band pass filters can be designed to be polarization-independent or polarization-dependent via cell alignment, and the bandwidth can be reversibly controlled from the original bandwidth 60–18 nm.

In addition to utilizing commonly used nematic liquid crystals and cholesteric liquid crystals, Chang li Sun et al. [71] proposed a near-infrared (NIR) filter using spherical phase liquid crystal (see Figure 11). It shows a low-operating electric field and large temperature gradient modulations. The central wavelength of the Bragg reflection can be thermally tuned from 1580 nm to 1324 nm with a temperature gradient of 42.7 nm/K. Meanwhile, the central wavelength can be electrically tuned over 76 nm within a low operating electric field of 0.3 V/μm. Thus, the SPLC filter may achieve a wavelength variation of 256 nm by thermal modulation and 76 nm by electrical modulation.

Similar to the tunable liquid crystal laser that will be discussed later, DBR has good optical filtering characteristics. Christophe Levallois et al. [72] combined the liquid crystals with photodetectors (Figure 12), achieving powerful spectral scanning in the wavelength range. The study of these tunable filters paves the way for the application of wavelength-tunable spectroscopic instruments in the visible and near-infrared response spectral ranges, revealing the great application potential of liquid crystal filters.

#### 3.2.2. THz Absorber and Filter Based on Meta-Material

LC is one of the few materials with high birefringence, low absorption loss and a large phase shift tuning range in the THz band. The most attractive property of LC is that its optical anisotropy can be controlled by changing the external electric field, magnetic field, optical field or temperature to prepare THz tunable devices that can achieve specific functions, such as THz switches, filters, absorbers and modulators. In recent years, the rise of metamaterials has provided new opportunities for the development of THz devices. Metamaterials are artificial structures with special electromagnetic properties formed by subwavelength metal or dielectric elements arranged in a specific manner. It can flexibly manipulate the light by designing the geometric structure to realize filters and absorbers. These metamaterials can be combined with liquid crystals to form a THz–LC tunable device.

The most classical structure is the Split-ring resonator (SRR), which can be regarded as a *LC* circuit [73,74], and its resonant frequency is determined by the Equation (5) [75]:(5)$$f=\frac{1}{2\pi \sqrt{LC}}\;\propto \;\frac{1}{n_{eff}}$$
L is the equivalent inductance determined by the enclosed area of the SRR, and C is the equivalent capacitance proportional to the dielectric constant of the medium deposited on the SRR. At the resonance frequency, certain frequency components of the signal pass, while other signals are highly damped. In order to describe the characteristics of the resonant circuit, the quality factor Q becomes an important parameter to measure the filtering performance of the resonant circuit.

Part of the research on THz–LC devices focuses on the design of metamaterial structures. Rafał Kowerdziej et al. [74] designed a subwavelength 25 nm-thick titanium split-ring resonator (SRR) planar square array to realize a tunable dual-band near-infrared metamaterial absorber using a combination of high-birefringence liquid crystals and high-loss metals (Figure 13a). Due to the high optical loss of Ti, perfect absorbance of over 99.4% was obtained at 328 THz and 364 THz. Furthermore, the absorption bandwidth is extended by the generated PSP resonance at the Ti-LC interface. The absorbance exceeds 70% over a broad spectral range of 318–372 THz, and spectral tunability up to 8 THz is achieved by simply switching the liquid crystal alignment wavelength. Shenghang Zhou et al. [76] proposed a tunable dielectric metamaterial absorber in the terahertz (THz) range. The absorber is composed of a silicon pillar array embedded in a liquid crystal (LC) layer, which is sandwiched by two graphene electrodes (Figure 13b,c). By way of varying the applied bias, the LC orientation can be continuously tuned. Structures such as single-layered fishnet metamaterial by Wei-Chih Wang [77] and keyhole-like metamaterial by Jun Yang [78] have also been proposed and tested.

Besides the study of the microstructure, Chia Yi Huang used the structures [73,79] to study how the resonance frequency influenced by the angles θ1 and θ2, polarized directions of the pump beam and LC directors with respect to the gaps of the SRRs (see Figure 14).

Similar to microwave and infrared light filters, the polarization sensitivity of THz filters/absorbers is also worth studying. Jun Yang and Deng Guangsheng et al. [80] designed a tunable polarization-dependent THz metamaterial absorber based on liquid crystals (LC) and studied the effect of the microstructure on TE and TM light (Figure 15). The absorption peak of TE polarized light is 239.5 GHz and that of TM polarized light is 306.6 GHz. By adjusting the bias voltage, the frequency tunability of the TE wave and TM wave are 4.7% and 4.1%, respectively.

### 3.3. Tunable Beam Controller

Beam control is also one of the common technical means in the optical field. Its main application is LIDAR [81,82,83], which is the cornerstone of various technologies, such as navigation, space communication, autonomous vehicles, drones and underwater vehicles. So far, many beam steering methods have been described [23,28]. Beam control devices involve beam deflection [29,84,85,86,87,88,89,90,91], optical switches [92,93,94], laser control and many other types. The traditional mechanical beam control device is relatively fixed and has the emphatic shortcomings such as a short service life, heavy weight, high power consumption and high cost. In contrast, LC-based beam steering devices can be lightweight, compact, low-powered and inexpensive. The following will select representative structures for elaboration. Other types of beam controllers are also briefly described below.

#### 3.3.1. Beam Deflection Control

The working principle of the LC-based beam deflection device is based on the phase modulation and diffraction. It mainly includes Bragg grating (Bragg diffraction), blazed grating (Raman-Nath diffraction), LC polarization diffraction grating (PB phase) and OPA (optical phased array). The concrete beam deflection theory was described by Wu Shin-Tson in [23]. The highly selective diffraction of blazed gratings can be achieved by the maximum retardation. Bragg gratings are similar to cholesteric liquid crystals (CLCs), which can vary the diffraction angle and wavelength by dynamically adjusting the pitch length. The PB phase is related to the evolution of light polarization. When the light beam passes through a nonuniform birefringent waveplate with local changes in the direction of the optical axis and uniform retardation, such as subwavelength polarization gratings, liquid crystal gratings and metasurfaces, the PB phase appears and relates to the rotation rate of the optical axis. Rowan Morris summarized many beam steering devices in [28], but some representative ones were not included, especially in LCPG.

According to previous studies on LCPG [13], a larger diffraction angle can be obtained by reducing its period ∧. However, for small period LCPGs that no longer satisfy the “thin” grating condition, the diffraction efficiency drops sharply. In the Raman–Nath region, despite a wide angular and spectral bandwidth, the gratings had a maximum single-order efficiency of less than 34% and produced many diffraction orders. Conversely in the Bragg region (Q > 10), when the incident wave propagated along the Bragg angle within the grating medium, single-order efficiencies of up to 100% were produced. In the case of linearly rotating the optical axis φ=πx/p, where p is the grating period, the PBD can theoretically deflect normal incident circularly polarized light toward first-order diffraction with an efficiency close to 100%. Combining the self-organized structure of a twisted liquid crystal with traditional PGs provided an important idea to solve the problem, which means that the Bragg diffraction and PB phase have become the current mainstream research directions. Both have a gradually rotating optical axis. At present, there are three main methods to achieve this by using optical alignment technology: a micro-projection exposure system based on a digital micromirror device (DMD), laser direct writing exposure system and interference exposure system. The most commonly used is the last one. It was mainly based on the interference of two orthogonal circularly polarized light beams. Theoretically, when two orthogonal circularly polarized light beams intersect at a certain angle, interference will occur, and the pattern of interference superposition is linearly polarized light with the periodic distribution of polarization direction rotation.

However, for traditional LCPG, the theoretical first-order diffraction efficiency of 100% can only be achieved if the half-wave condition is met, and zero order will appear if the half-wave condition is not met. Therefore, the traditional LCPG can only achieve 100% of the first-order diffraction efficiency of a single wavelength and cannot maintain a high diffraction efficiency in a wide band range. Since it is difficult for traditional LCPGs to maintain the high diffraction efficiency in a wide band, in order to solve the dispersion problem, a multilayer twist structure has been proposed. Xiao Xiang et al. [84] introduced transmissive Bragg PGs based on single-layer and double-layer twisted structures with large diffraction angles by creating two or more tilts within the grating (Figure 16). The fabrication of tilted gratings was also mentioned in their experiments. The chiral dopant-induced helical twist can effectively generate tilted gratings, and it was verified that the tilt angle θ of the grating is determined by the three parameters of twist angle, thickness and surface period. Compared with the nanoscale-tilted Bragg grating in 2017 [95], which FOV was 19–29° at 335 nm with a 450 nm grating period, the double-layer structure experimented in 2018 improved a lot, showing a 40° angular bandwidth, 200 nm spectral bandwidth, 76% average efficiency and 96% peak efficiency at 532 nm with a 400 nm grating period. This idea is based on the introduction of a new controllable variable in the original parameters of LCPG to eliminate the influence of wavelength change and the use of positive and negative twisted symmetrical structures to compensate for the changes in the diffraction efficiency caused by the introduction of the distortion.

In 2019, Kun-Yin et al. [90] proposed to attach PVG to a flexible thin film to make it stretchable, and rollable 3 µm-thick PVGs were prepared on a 160 µm-thick polydimethylsiloxane (PDMS) substrate. By mechanical stretching, the horizontal direction of PVG is elongated, and the total thickness is reduced (Figure 17a,b). The Bragg period value changes, which affects the reflection band and deflection angle. Experiments show that the central wavelength is blue-shifted from 507.5 nm to 474.5 nm, and the deflection angle can be tuned from 35° to 46.5° with a tunable range of 11.5°. This study shows the potential for laser beam control applications and support the fabrication of flexible devices.

Since PB diffraction is highly polarization-selective, the most commonly used passive scheme is to control the input polarization state to switch the deflection angle of the PBD so that the output beam can switch between +1 and −1 diffraction orders. For active PBDs made of liquid crystals, the input beam can be switched between zero and first-order diffraction orders simply by applying a voltage to the PBDs, similar to a variable step blazed grating. The fabrication of PBD employs polarization holography, which transfers the linearly rotated polarization pattern to the LC director. In 2020, Comrun Yousefzadeh et al. [87]. proposed a nonmechanical beam steering device based on the Pancharatnam–Berry phase in liquid crystals, using a linear phase control elements (PCEs) array to locally control the orientation of the liquid crystal director. The device model is optimized to achieve a maximum deflection of 1.7° and a diffraction efficiency of 90%.

Inspired by the PB phase and multi-twisted structure, in 2020, Junyu Zou et al. [85] designed and experimentally demonstrated an efficient achromatic wide-angle PBD, which consisted of a three-layer mixture with liquid crystal thicknesses of 0.9 µm, 1.32 µm and 0.88 µm each (see Figure 17c), and proposed a new method that can precisely measure the thickness and twist angle of liquid crystal films, measuring the corresponding twist angles of each layer as −69.4°, 3.7° and 64°. The deflection angles at 457 nm, 532 nm and 635 nm were 0.75°, 0.87° and 1°, respectively, and the first-order diffraction efficiency could reach almost 100% under normal incidence. The study of such long-period polarization-dependent gratings is very useful for the resolution enhancement of near-eye displays.

The above studies were all based on the vicinity of visible light. As the research progresses to THz, the scheme of realizing the periodic distribution of the refractive index using the programmable element surface to realize the diffraction grating has been developed. By the binary encoding of subsurface units with different phase or amplitude responses and designing the encoding patterns, far-field scattered beams can be flexibly controlled, including single-beam, dual-beam and multi-beam radiation; polarization-dependent beam steering; beam steering and more. This is the specific use of the OPA (optical phased array) method and has been experimentally demonstrated that OPA is suitable for THZ deflection. Programmable metasurfaces enable novel communication and imaging systems.

In 2020, based on the pixelization of THz devices (Figure 18a,b), Jingbo Wu et al. [29] applied the electrical addressing technology of a liquid crystal display and used terahertz programmable metasurfaces to achieve terahertz beam deflection, with a maximum deflection angle of 32°. Its principle is similar to the periodic phased array structure. The change of the periodic phase distribution affects the deflection angle, which is determined by Equation (6) [29]:(6)$$\left|\sin\theta_{s}-\sin\theta_{o}\right|=\frac{\lambda }{Md}.$$
where M: the number of elements of the periodic structure, d: the distance between elements and *θ_s_* and *θ_o_* incident angle and exit angle. After continuous coding optimization, it can achieve 32° beam deflection at 672 GHz. Based on similar ideas, Yanchun Shen et al. [91] applied optical alignment technology to form a periodic gradient distribution of LC. The frequency-independent geometric phase modulation can be obtained to form a blazed grating, which realizes light deflection of 13–19° in the 0.8–1.2 THz.

Wide angle is also an important measure of the performance of beam deflection devices. Xiao Jian Fu et al. [86] introduced different metal resonance structures (see Figure 19a) in the LC tunable layer and used generalized Snell’s law to generate coding sequences and assign the coding sequences to reflective units. Beam deflection at large angles of 20–60° in the range of 0.630–0.650 THz.

The above structure realizes a terahertz programmable metasurface in reflection mode, and can be realized for transmitted light. The new metasurface structure proposed by Prof. Xiao Jian Fu’s [89] team achieves transmission multi-beam deflection at 0.426 THz with a maximum deflection angle of 30° (see Figure 19b). It should be noted that in order to evaluate whether the complex reflectivity satisfies the requirements for 0 and 1 elements of a 1-bit encoded metasurface, a parameter Binary Bit Reflectivity Ratio (BBRR) is defined. *T*_0_ and *T*_1_ are the complex reflections in the two states, and BBRR is the ratio of the moduli of the difference between *T*_0_ and *T*_1_ to the moduli of their sum. Ideally, *T*_0_ and *T*_1_ should have the same amplitude and 180° phase difference, and the BBRR is infinite. The higher the BBRR, the higher the diffraction efficiency. The device can be designed as a reflective or transmissive according to the BBRR as Equation (7) explain [29]:(7)$$BBRR=\left|\frac{T_{0}-T_{1}}{T_{0}+T}\right|.$$

#### 3.3.2. Laser Polarization Mode

Tunable vertical cavity surface emitting lasers (VCSELs) have the characteristics of low threshold current and single-mode continuous wavelength tuning characteristics, and have broad application prospects in the fields of optical communication, optical storage, spectroscopy and medical treatment. However, traditional VCSELs have weak anisotropy between polarization modes due to their symmetrical waveguide structure, and it is difficult to achieve stable single-polarization mode output. When the operating temperature and injection current of the device change, the VCSEL will switch between polarization modes. In 2006, Levalloisa et al. [96,97] first proposed to control the output polarization mode of VCSEL by embedding the liquid crystal layer structure in the cavity and use the birefringence to increase the threshold gain difference between the polarization modes in the cavity, and finally, the VCSEL can output stable single polarization mode light. At the same time, liquid crystal, as an electro-optical refractive index modulation material, can continuously tune the output wavelength of VCSEL by changing the equivalent refractive index of e-light under voltage.

The main team currently researching liquid crystal tunable lasers is Zou Yonggang group from Changchun University of Science and Technology and C. Levallois from Univ Rennes. Many works have been done on the basic structure [98]. In 2018, an intra-cavity liquid crystal tunable VCSEL was designed with two parts [99], whose central wavelength of 852 nm, a liquid crystal micronanocavity controlling the light phase and a half-VCSEL structure, including a high reflectivity dielectric Bragg mirror and an active 852 nm light-emitting region. The relationship between the applied tuning voltage and the threshold gain of the laser with different polarizations was measured experimentally. The threshold gains of the o-polarized light remained unchanged, while the threshold gains of the e-polarized light first decreased and then increased. It makes e-light more competitive in lasing and realizes single-polarization lasing. In the experiment, the continuous wavelength tuning range of 31 nm and the stable output of laser single polarization were realized, and the tuning efficiency was greater than 10 nm/V. Based on the original structure, Zou Yonggang optimized the coupling layer structure between [98] the semiconductor cavity and the liquid crystal cavity to make the liquid crystal tunable VCSEL have a wider wavelength tuning range, up to 41.1 nm. In 2020, C. Levallois et al. [100] investigated an InP-based VCSEL with a liquid crystal (LC) microcell monolithically integrated. This tunable VCSEL operates in CW at room temperature and exhibits more than 23 nm wavelength tuning around 1.55 μm at a maximum applied voltage of 20 V. The measured laser threshold around 6.5 mW is still comparable to VCSEL without LC microcell, a clear indication that the optical losses related the LC are very low.

In recent experimental studies, both DBR and HCG can meet lasing conditions and achieve ultra-broadband and high reflectivity. DBR is a multi-layer film structure, and HCG is a sub-wavelength grating structure in which a high-refractive-index material is completely surrounded by a low-refractive-index insulating dielectric (such as oxide or air) as showed in Figure 20. Using HCG, whose thickness is about a few tenths of DBR, as the upper mirror, can reduce the epitaxial thickness, reduce the difficulty of material growth, and simultaneously control the polarization of the emitted light and provide optical feedback. Compared with DBR, HCG is easier to embed liquid crystal and HCG simplifies the fabrication of devices, and is an ideal structure for the mirror in liquid crystal tunable VCSELs. Jing Zhang et al. [101] experimentally demonstrate for the first time to our knowledge electrically injected vertical-cavity surface-emitting lasers (VCSELs) with post-supported high-contrast gratings (HCGs) at 940 nm. Combined with theory that the HCG-VCSEL with a *λ*/2-cavity for the transverse magnetic polarization has a smaller effective mode length of 1.38·(*λ*/*n*), the −3 dB frequency of the HCG-VCSEL can theoretically reach 46.8 GHz at 12 mA.

Although beam deflectors and polarization mode laser output controller can change the light, the difference is changing from the inside or outside. Both provide a reference for the application of liquid crystal in the field of laser control.

#### 3.3.3. Optical Switch

Optical switching is the most basic optical component in optical field and has wide applications, but there are relatively few studies based on liquid crystal optical switches in recent years. Optical switch controllers can be divided into two categories: one is the spectral optical switch controller, and the other is the channel optical switch controller. The spectral optical switch controller can be equivalent to the optical filter and optical absorber, which is used to control the transmission of light at a specific wavelength. Huan Peng Xin et al. [92] introduced localized surface plasmon resonance into the liquid crystal layer, designed a unique polarization-dependent au nanorod unit structure (see Figure 21a), and used the coupling of nanostructured particles to form interference phenomena such as Fano resonance to achieve optical tuning in the near-infrared region. Since the Fano resonance is very sensitive to the changes of the local environment, the external liquid crystal can realize tunable control, and the optical switching depth of 70% in 750 nm–1450 nm is realized.

In the channel light optical switching control, Jose M. Otto et al. [93] combined liquid crystal with MMI, and used the refractive index of LC to tune the effective distance between the waveguides(see Figure 21b). Optical functions such as variable coupling and optical switching have been demonstrated. In this work, liquid crystals serve as electrooptically active cladding to drive integrated waveguides, providing possibility for LC to be applied in optical integrated circuits.

### 3.4. Bistable State LC Device in Livelihood

The devices referred to above are still in the research stage and far from commercial, and several mature liquid crystal technologies have appeared on the market and have been applied to our daily life since its good switch ability, such as electronic paper (e-paper) [102,103,104,105,106,107] and smart windows [108,109,110,111,112,113,114], especially the field related to the display, which has developed for nearly two decades.

#### 3.4.1. E-Paper

With the strengthening of environmental awareness, the emergence of e-paper has successfully promoted the environmental trend of paperless office. Liquid crystals used for the e-paper include TNLC [106] (twist nematic LC) and PDLC [103] (polymer Dispersed), the potential of CLC has been demonstrated in [107]. Due to different LC characteristics, the corresponding working principles are slightly different. TNLC, the same as the CLC, LC can be rotated by electric field. Applying a voltage to the two electrodes will cause the polarization of the LC to turn parallel to the electric field direction. The deflection angle determines the excessive luminous flux, which is also one of the most common technologies of LCD.

The preparation process of PDLC materials is usually to uniformly disperse low molecular nematic liquid crystals in transparent polymers, induce phase separation through photo-polymerization, thermal initiation or solvent volatilization and other methods, so that liquid crystals can precipitate from the polymer to form droplets, and uniformly disperse in the polymer network. When no voltage is applied, the ordinary light refractive index of the liquid crystal droplet is mismatched with the refractive index of the polymer with light scattering; when a certain voltage is applied, the LC is aligned along the electric field with light transmission [103].The research of e-paper is mainly divided into two directions: one is electronic paper optical alignment technology and especially focus on the limitation of exposure time, which has been summarized in [102,104]; another one is to make the displayer more functional and stretchable [105,106].

#### 3.4.2. Smart Windows

Similar to the e-paper, smart windows are also designed for smart building and environmental protection. in order to meet the energy saving needs of buildings, tunable smart windows are also a research hotspot.

LC used for smart window includes PDLC, PSLC and CLC, which also determined the performance of the smart windows. Hemaida et al. [111,114] prepared a PDLC smart window with strong scattering effect [103], which realized the electronically transparency control. Based on the PDLC, newly material PSLC also was demonstrated in smart window [109]. CLC with appropriate pitch and average refractive index value can reflect the infrared ray, so it is suitable for infrared reflectors. Xiaoxue Du et al. [110] used bistable dual frequency CLC to realize 700–1400 nm infrared reflection. NIR rang 1050–1550 nm [112] and 800–2500 nm [113] are also be achieved successively. The transparency adjustable smart window is based on scattering and cannot completely isolate light. So maybe the TNLC type smart windows can be made, such as e-paper.

The optically smart equipment based on LC brings us convenience. Dissimilar to the equipment in the three subsections above, it is difficult to industrialize with multi complex structures. It is hoped that the improvement of LC devices includes not only performance, but also industrialization and convenience for future.

## 4. Optofluidic Devices Based on Conventional Liquids

The technologies used in conventional liquid lens can be divided into two categories: interface control and diffusion distribution. The former can control the shape of the interface and has certain limitations, so the main research scope is lens. However, the optical devices based on the diffusion have better flexibility in adjusting the refractive index distribution with a wider application. It can also combine with transform optics to realize waveguide bending and GRIN resonators.

### 4.1. Liquid Lens

Since the liquid lens has developed for many years, there are few innovations in the interface control principle in current research, mainly focusing on the design of the structure or the optimization of the interface. Next, we will introduce the liquid lens according to the working principle of the lens.

#### 4.1.1. Electrically Tunable Lenses

Leihao Chen et al. [115] reviewed the electrically tunable lenses in 2021, which mainly focuses on the development and unique taxonomy of liquid lenses based on conventional liquid and DE film in nearly 20 years. However, the LC lens, which is almost ignored in this paper, has been explained in more detail in above. In this section, the representative structures of the electrically tunable lens in the past five years will be selected for analysis.

Electrical control is widely used in adaptive liquid lenses, including electro wetting (EW), liquid dielectrophoresis (LDEP) [41] and the dielectric elastomer lens [116,117,118,119,120] in recent years. Electrowetting lens has been studied for nearly two decades, mainly focusing on the single electrode EW lens, which has led to a gradual decline in the research interest. Nowadays, it is more inclined to the multielectrode electrowetting lens and the improvement of imaging quality and related application. The research on LDEP in liquid lenses has just been developed, mainly involving the structures design based on its unique advantages. At present, the main teams to study LDEP lens were the Qingming Chen group and Xumiao group [121,122]. Additionally, dielectric elastomers are gradually entering the researcher’s vision.

Kartikeya Mishra et al. [123] proposed a novel electronically adaptive optofluidic lens allowing to manipulation the focal length and asphericity, consisting of two parts: adjusting the average curvature by electrowetting and limiting the spherical aberration through local Maxwell force (see Figure 22). The device is fully encapsulated and contains a liquid lens with a transparent aperture of 2 mm immersed in ambient oil. The focal length of the device can be tuned from 10–30 mm. The chamber structure of the liquid lens is generally cylindrical [124] or conical [125], and the filling liquids generally contain water. Chao Liu et al. [126] proposed a non-aqueous organic solution based large-aperture spherical electrowetting liquid lens (SELL) with a wide tunable focal length rang. The spherical chamber can enlarge the contact angle range and the focal length range compared with the conventional cylindrical electrowetting liquid lens. The SELL has a low threshold voltage, high breakdown voltage, fast response time and large zoom ratio with excellent image quality, and it has been experimentally proven that the non-aqueous organic liquids can avoid the generation of bubbles resulting from the hydrolysis phenomenon and still keep the lens working even when the dielectric layer fails. At a 1:1 volume ratio of conductive liquid to insulating liquid, the focal length ranges from (−∞, −80.4 mm) to (0.3 mm, +∞). Furthermore, the focal length range can be expanded under different volume ratios of a conductive liquid to insulating liquid.

Furthermore, Chao Liu et al. [127] tried to combine single and multi-electrode to electrowetting lens (see Figure 23). The upper and lower cavity voltages were controlled separately to realize the focusing and deflection of the beam. The focal length is tuned from −180 mm to −∞ and +∞ to 161 mm, and the beam deflection angle can be adjusted from 0° to 22.8°.

Besides the device design, Jaebum Park [128] conducted studies on the voltage applied to EW devices. Although the AC (alternating current) voltage is often applied to an electrode in EWOD devices to improve electrical characteristics (see Figure 24), it may cause oscillations of liquid interfaces that can be detrimental to the system’s performance as an optical device. The origin of the interfacial oscillations of polymeric electrolyte solutions and the frequency condition to eliminate such ripples are identified separately by observing the dynamic responses of contact lines as a function of the AC frequency, and by measuring the relaxation time scale, it is possible to find the key AC frequency to ensure stable interface control, which further improves the development of EW devices.

The research on DEP appeared relatively early, but it is first applied to the biological field for material separation. Since DEP is applied to liquid lens, Wu Shin-tong and Qingming Chen engaged in the design of the DEP device in recent years. In 2018, Qingming Chen et al. [129] proposed a novel in-plane optical jet lens driven by DEP force (Figure 25a). In an open microfluidic channel, a thin layer of silicone oil is sandwiched between two glasses, one has a patterned straight strip as the top electrode and the other has a uniform ITO layer as the bottom electrode. Initially, a concave liquid–air interface is formed at the end of the open microchannel by capillary flow. Then, the electric field exerts a net DEP force to continuously modify the liquid–air interface, as well as the focal length. When the driving voltage increases from 0 V to 260 V, the focal length changes from approximately −1 mm to −∞ and then from +1 mm to +∞. In particular, longitudinal spherical aberration (LSA) is effectively suppressed when the lens works at the focusing state, making LSA < 0.04. at the same year, Qingming Chen extended the DEP lens from single-side control to two-side control [130], the DEP force changed the air–liquid interface from biconcave to biconvex, and the focal length tuning range was expanded synchronously (Figure 25b).

Aberration is a long-standing problem of fixed focal lenses, and a complicated lens set is usually required to compensate for aberration. In 2020, Qingming Chen reshape the interface to compensate the aberration through the DEP [131]. The key idea is to use two arrays of electrode strips to symmetrically control the two air–liquid interfaces by the dielectrophoretic effect (Figure 26). The strips work together to define the global shape of the lens interface and, thus, the focal length, whereas each strip regulates the local curvature of the interface to focus the paraxial and peripheral arrays on the same point. Experiments using a silicone oil droplet demonstrate the tuning of focal length over 500–1400 μm and obtain a LSA of ~3.5 μm, which is only 1/24 of the LSA (85 μm) of the spherical lens. Fine adjustment of the applied voltages of strips allows even elimination of the LSA and enabling of the aberration-free tunable lenses.

The research on dielectric elastomer (DE) lens has already developed for a decade. Initially, the DE membrane lenses’ tunability relays on the stiff electrodes [117], the compliant electrodes [120] and the gel electrode [119], which deforms when applying the voltage. The research based on the three electrodes is still ongoing. Wang et al. coupled the stiff electrode to a soft elastomeric membrane [117], and the attraction between electrodes causes a bulging of the elastomeric material in the central part, generating a biconvex lens. Yang Cheng et al. proposed a varifocal liquid lens driven by a conical dielectric [120], in the activation state, the frame upwards when an actuation voltage is applied to the compliant electrodes of the bottom conical dielectric elastomer, causing the focal length of the liquid lens changed. Liu et al. developed a new self-contained focus-tunable lenses based on transparent and conductive gels [119], when a voltage is applied to the gel lenses, the elastomer membrane is stretched radially by Maxwell force, resulting in a reduction of lens thickness that corresponds to an increase of curvature radius.

In 2021, Yang Cheng et al. [116] propose and demonstrate a compact tunable lens with high transmittance using a DE sandwiched by transparent conductive liquid(see Figure 27). An active membrane determines the focal lens while the applied voltage off (Figure 27a) or on (Figure 27b). Actually this structure evolved from the structure explained by Shian [132], the upgrade exhibits greater tunability and compactness. The transparent conductive liquid not only serves as the refractive material of the tunable lens but also works as the compliant electrode of the dielectric elastomer. The overall dimensions of the tunable lens are 16 mm in diameter and 10 mm in height, the optical transmittance at 380–760 nm is as high as 92.2%, and the optical power change of the tunable lens is −23.71 D at an actuation voltage of 3.0 kV.

Based on this model, the effects of a set of parameters, including chamber radii, shear modulus, permittivity, pre-stretch ratios and injected liquid volumes, on the tuning performance of the lens were analyzed by Chi Zhang [118]. It was found that, by regulating the liquid volume in each chamber, both the initial focal length and the tuning range could be adjusted easily. Under the condition with specific liquid volumes, the lens possesses both positive and negative focal length during voltage actuation, indicating promoted tuning performance, which is acclaimed for optimal design.

#### 4.1.2. Thermal Effect Liquid Lens

A. Yu. Malyuk, a team focusing on thermal effect liquid lens research, proposed a simple and easy-to-implement varifocal liquid lens structure driven by laser-induced thermal Marangoni forces [133]. The lens consists of low volatility droplets placed on a transparent solid surface. The focal length tunability is achieved by changing the local curvature of the droplet surface, which is caused by the thermocapillary displacement of the liquid from the core part of the droplet to its edge due to laser heating. Depending on the power of the laser beam, the droplets can act as either a varifocal condenser lens or a varifocal diverging lens. In contrast to most analogies, the manufacture of this lens does not require expensive materials or liquids, special structures or high-tech machining. The droplet can behave as a planoconvex or planoconcave lens and can be effectively adjusted to vary between two modes (Figure 28a).

A. Yu. Malyuk et al. [134] also demonstrated an adaptive liquid lens controlled by laser-induced solutocapillary forces. The liquid is a binary mixture of ethanol and ethylene glycol, and the requirement for the liquid selection is high surface tension. The mixture contains endothermic dyes, and the localized heating by the laser beam results in a local decrease in the concentration of the volatile liquid. Finally, axisymmetric temperature gradient and concentration gradient appeared on the free surface of the mixed layer (Figure 28b). Since surface tension is jointly determined by temperature and concentration, then their gradients lead to surface tension gradients along the interface, causing liquid surface deformation. By varying the intensity of the laser beam, the droplet’s shape, aperture and focal length can be reversibly changed without delay. Research on laser-induced lenses will facilitate the development of smart liquid optical devices that simulate the accommodative reflex and pupillary light reflex of the eye.

#### 4.1.3. Heat and Concentration Diffusion GRIN Lens

The optofluidic devices based on the diffusion convection equation mainly include thermal diffusion and molecular diffusion. The research on the solute diffusion has gradually turned to the fluid mixing structure; In terms of thermal diffusion, the device structure design is preferred to produce more perfect thermal distribution. In recent years, the research of such devices has been relatively reduced.

The diffusion process is unique, which cannot be found in solid devices. More specifically, liquid diffusion generates a concentration gradient, namely forming a refractive index gradient. Miscible liquids and their mutual diffusion are of great significance in the design of optical fluid devices, and their applications in beam shifters, curved waveguides, beam splitters and Luneburg lens have been revealed. The grin distribution in the jet system is adjusted by changing the flow parameters (such as *Pe*) or changing different types of liquids with different refractive indices and diffusion coefficients [45]. Grin modulation provides great flexibility and adjustability for optical operation of optical waveguide system. The classic model of miscible liquid is shown in Figure 29a,b and corresponds to its equivalent optical path model [135].

The refractive index of a liquid is always related to its temperature. In micro-channels with different temperatures, the temperature distribution through thermal conduction or thermal diffusion will produce a gradient refractive index distribution similar to mass diffusion. Compared with the traditional mass diffusion, the thermal diffusion can form a relatively uniform gradient refractive index distribution in the uniform liquid flow. At the same time, utilizing single liquid is convenient for the recycling of the liquid, which conforms to the environmental protection.

Zhang et al. [136] demonstrated an optofluidic lens based on a laser-induced thermal gradient (see Figure 30). This scheme is achieved with an additional light field, where two straight chrome strips are made at the bottom of the channel in order to absorb the pump laser. Benzyl alcohol solution is applied in liquid thermal lens, since a larger refractive index change can be obtained compared with other liquids under a certain temperature difference. A 2D refractive index gradient will form between the two metal strips. The results show that the focal length can be continuously tuned from infinity to 1.3 mm. Meanwhile, off-axis focusing can be achieved by compensating for the hot spot of the pump laser. This tunable lens has the advantages of small size, easy integration and fast response.

Liu et al. [137] also reported a liquid thermal gradient index lens in a homogeneous fluid (see Figure 31). The trapezoidal region in the upper layer was used to establish a gradient index profile through thermal conduction between three streams of benzyl alcohol with different temperatures. A compensation liquid is added to the rhombus region of the lower layer to form a stable square law parabolic refractive index profile only in the lateral direction. The focal length of the thermal lens can be adjusted from 500 µm to 430 µm. In this design, a high enhancement factor of 5.4 can be achieved, the full-width-at-half-maximum of the laser is 4 µm, and the response time of GRIN lens is about 20 ms.

### 4.2. Other Optical Liquid Device

Although other liquid optical devices are not as widely studied as lenses, they have been developed to a certain extent, including curved waveguides [43,44], ring resonators [138], tunable apertures [139,140], optical switches [141] and phase controllers [142].

Optical bending waveguide is an indispensable part of integrated optical system, so many methods to reduce bending loss are proposed. In a transformation optical bending device, the light travels along the curve without reflections as if it was propagating in a straight one, which results in light propagation with invariable intensity profile and no bend loss regardless of bending angles. So far, those optical bending devices are still solid-based and lack reconfiguration once fabricated. Liquids own the ability to form tunable refractive-index gradient, showing the potential to be applied in TO (transform optical) devices such as tunable liquid-based waveguides and cloaks. Yi Yang et al. proposed two different schemes to realize waveguide bending in 2017 [43] and 2020 [44], the concentration diffusion based on counter flow (Figure 32a), and the thermal diffusion based on constant temperature boundary is shown in Figure 32b.

In view of the advantages of optical fluid diffusion, Yi Yang’s group introduced the gradient refractive index profile formed by liquid diffusion in the annular channel into the annular resonator to form a tunable unidirectional emission gradient refractive index resonator in 2020 [138] (see Figure 33). In the simulation and experiment, the influence of different bending radii on the optical characteristics of the resonator is studied. The results show that the device with a bending radius of 100 μm has a squeezed light coefficient of about 1.8 (see Figure 33b,c), and the divergence angle can be as small as 14°, which can be used for future laser emission. Through continuous research on novel liquid gradient refractive index devices, the Yi Yang group has profoundly revealed the great application potential of diffusive liquid devices.

The sensor diaphragm is also an important element in the optical system, and the image quality can be improved by adjusting the aperture. Pressure actuation is the main way, but it cannot precisely control the aperture and has a long response time, so the research of diaphragm is mainly focused on other directions. present a laser-actuated adaptive optical diaphragm that is capable of aligning the disturbance of the coaxiality of the optical signal and the plane of aperture. The diaphragm consists of two layers of immiscible liquids (see Figure 34), where the bottom layer absorbs a pumping laser beam and transmits an optical signal, while the upper layer transmits the pumping laser beam and stops the optical signal [139]. The operating principle is based on creating the circular thermocapillary rupture of the upper layer by Marangoni forces induced by heating with the pumping laser beam. The thermocapillary rupture serves as an aperture of the diaphragm. The aperture diameter at a fixed power of the laser beam depends on the upper layer thickness and reaches diameters up to two times larger in comparison with diaphragms operating on electrowetting and dielectrophoresis. The aperture tuning ratio is 100%. By shifting the pumping laser beam in the plane of the diaphragm, the aperture can be displaced for a distance up to a few of its radii.

In addition, Tao Chen et al. [142] designed a phase modulator based on electrowetting. Figure 35a shows the cross section of the optofluidic phase modulator, which is mainly composed of an upper cover sheet, a lower cover sheet, an outer circular cavity, an inner cylindrical cavity and a transparent sheet. In order to facilitate the flow of liquid in the chamber, there are four circular holes on the inner cylindrical chamber, and the height of the inner chamber is slightly shorter than that of the outer chamber. An insulating droplet is deposited in the center of the bottom, which is surrounded by a conductive liquid. The inner cavity is an effective part of the optical phase modulation, and the function of the outer cavity is to store the liquid. According to the electrowetting effect, when a voltage is applied, the wettability of the conductive liquid on the solid surface changes, so the shape of the insulating droplet also changes, resulting in a variable optical path length, thereby realizing the phase adjustment function. Indium tin oxide (ITO) planar electrode is used in the optofluidic phase modulator. Two electrode structures are presented, namely key type and interdigital type (see Figure 35b,c).

Additionally, the interdigital type electrodes can better control the planar movement of liquid and be applied in many studies, as in [122,143]. The experimental results also show that the two electrodes can achieve 6.68 π and 9.366 π, respectively, at the same voltage of 150 V. This research has great application potential in adaptive optics of coherence tomography and phase shift interferometry.

## 5. Conclusions and the Future

Based on the advantages of tunability, liquid devices are valued and applied in many fields, including augmented reality, holographic imaging, etc. Table 1 lists the classical references of LC devices, including recent typical structures and high-quality reviews before 2018. Due to the isotropic liquid unique characteristics, Table 2 lists another structure.

However, the challenges are increasingly notable. Although there are many kinds of liquid devices with different research points, the response time is a common difficulty to overcome. The interface deformation of ordinary liquid is deeply affected, and the response time reaches to seconds-order, which is mainly caused by the vibration at the interface. Although the diffusion method has greatly improved the modulation speed comparable to that of LC devices, it is still a sub-second response time, which means that tunable devices are difficult to meet the needs of beam direction switching in radar systems and adaptive lenses imaging in real time detection.

Due to the popular research of adaptive lens, different types of liquid lens have been demonstrated, which provides a prospect in precision medical, ultraportable image recorders and other equipment for their small size and light weight. At present, the focal range and aperture of the ordinary liquid lens are slightly better than those of the LC lens, but the aberration and applied voltage are on the contrary, which means the liquid lenses can still be improved. For the ordinary liquid lens, the near spherical interface will cause longitudinal spherical aberration, which is harmful to the focusing of a light spot. The diffusion lens needs continuous liquid supply, which hinders the miniaturization of devices. For LC devices, the major problem is that the constant refractive index difference of LC materials limits the simultaneous improvement of the aperture and focal range; another challenge needs to be solved is the high polarization dependence of LC lens, as the existing research on polarization independence cannot meet the commercial standard. At the same time, the reflection of the multilayer interface of the device needs to be considered by both lens, which may introduce a high loss of intensity. These challenges may be solved through the single layer joint control of and multiple regions (radial division), similar to the programmable LC deflector but involve the harsh selection and matching of LCs in each region.

Due to the isotropy of the ordinary liquid, it depends on the control means, which limits the diversity of the device. Hybridization may be the only way to develop. However, conversely, the birefringence and the diversity of its structure, LC can realize more optical devices, including THz LC devices, beam deflection control and polarization mode laser output. In THz band, the high absorption of THz by ordinary liquid, such as water, limits its application in this band, so the research is relatively less. The outstanding characteristics of LC in THz band make THz LC devices gradually become a research hotspot. Although most devices show exciting tuning performance at present, the research on THz LC devices has just started. There are challenges, such as low tuning speed, narrow phase modulation range and insufficient accuracy. The possible reason may be that the traditional LC physics theory and technologies are no longer applicable at the submillimeter scale. Therefore, the research of THz LC devices can be developed in the following directions: firstly, deeply exploring the interaction between LC and THz wave, the external field regulation and surface interaction of liquid crystal at the THz wavelength scale have become urgent problems to be solved; secondly, improving terahertz modulation technology and developing a new generation of terahertz devices with high modulation depth, high switching rate, low consumption and low cost. The unit structures of THz presented is lesser. It may be a feasible scheme to summarize and study the mechanisms by drawing the nanostructures from other fields such as nanowire arrays.

In beam steering, ordinary liquid devices can only rely on the refraction of the interface, but the flatness is not ideal. The natural advantages of LC material with unique stratified and periodic spiral structure make it possible to achieve beam steering by diffraction. However, due to the limitations of itself, ideal phase modulation may not be obtained under the applied voltage, resulting in low deflection efficiency and low accuracy, especially in the case of two-dimension scanning, which will be the key point to be solved. Specifically, for the LCPG, it can only achieve high diffraction efficiency when circularly polarized light incident, but the diffraction efficiency of linearly polarized incident beam drops to 50%. Therefore, it is necessary to break through the limit of the incident polarization state of LCPG, such as the high diffraction efficiency grating device structure for linearly polarized light modulation. The most urgent issue of deflection efficiency and angle may be realized by device hybridization as well. After all, PVG and PBD theory have 100% diffraction efficiency.

The tunability of liquid devices has a great impact on optical research. We hope that this review can help scholars handle the recent development of liquid devices and provide a reference for the research liquid devices in the future.

## Figures and Tables

**Figure 1 molecules-27-08025-f001:**
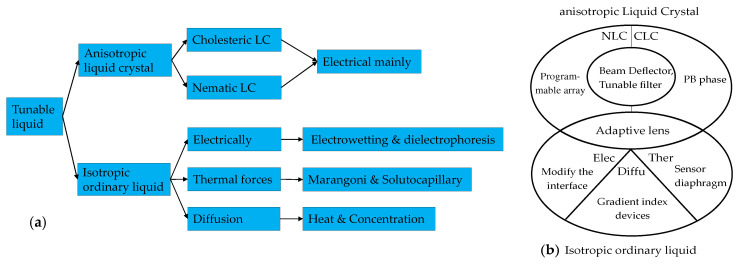
A brief introduction of the tunable liquid devices’ (**a**) working principle and (**b**) a diagram that presents the common and unique device type.

**Figure 2 molecules-27-08025-f002:**
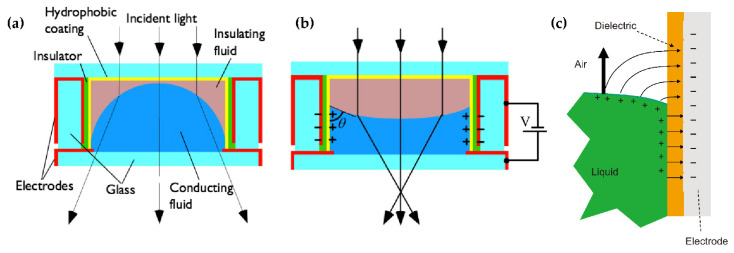
Electrical control diagrams of (**a**) electrowetting when the voltage is off (**b**) and the voltage is on [40]. (**c**) The diagrams of dielectrophoresis [41]. Reprinted with permission from Refs. [40,41]. 2004 AIP Publishing and 2013 Elsevier Science.

**Figure 3 molecules-27-08025-f003:**
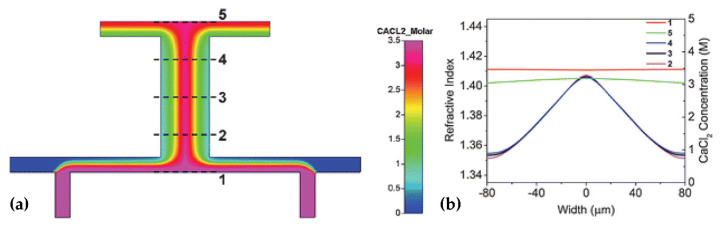
Simulated refractive index profile due to the diffusion [42]. (**a**) Simulated model and distribution graph. (**b**) Cross-sectional refractive index distribution at different locations along the flow direction (1, 2, 3, 4 and 5, as indicated in (**a**)). Reprinted with permission from Ref. [42]. 2009 Royal Society of Chemistry.

**Figure 4 molecules-27-08025-f004:**
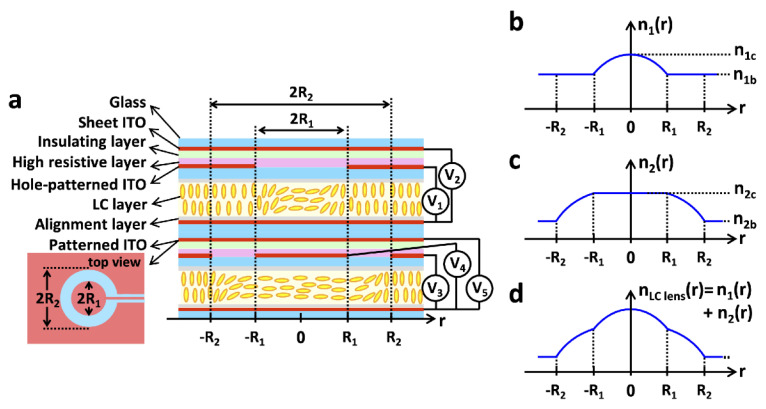
NLC spatially expanded phase distribution. (**a**) Schematic diagram of the structure of the proposed LC lens. (**b**) The upper LC refractive index varies with the radius. (**c**) Lower LC refractive index as a function of radius. (**d**) Sum of the refractive indices of the upper and lower layers [47]. Reprinted with permission from Ref. [47]. 2019 The Optical Society.

**Figure 5 molecules-27-08025-f005:**
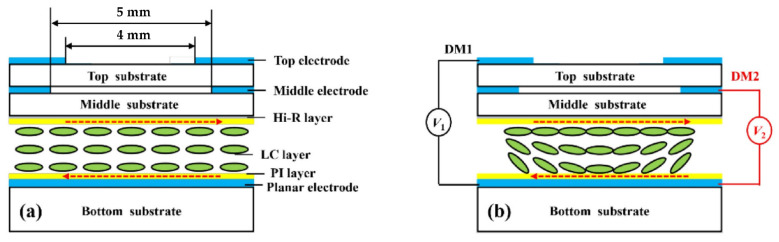
Dual-aperture LC lens at different working states: (**a**) when the voltage is on and (**b**) the voltage is off [49]. Reprinted with permission from Ref. [49]. 2022 Elsevier.

**Figure 6 molecules-27-08025-f006:**
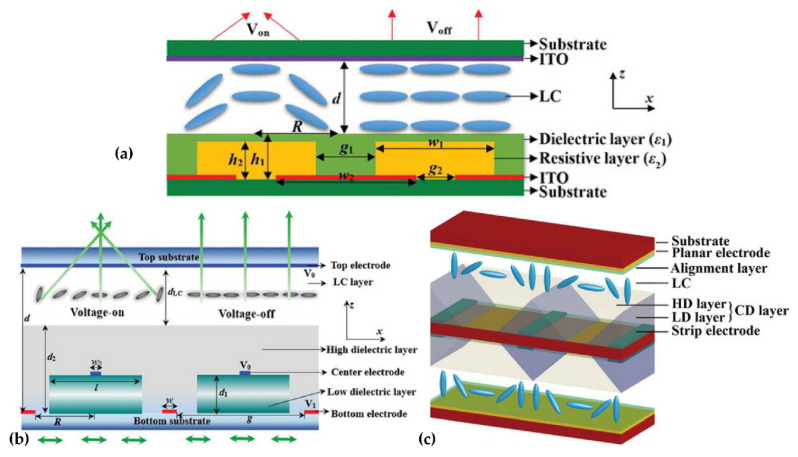
LC lens arrays with different layers: (**a**) resistive layer [27], (**b**) low dielectric layer [50] and (**c**) composited dielectric layer [51]. Reprinted with permission from Refs. [27,50,51]. 2020 Taylor & Francis.

**Figure 7 molecules-27-08025-f007:**
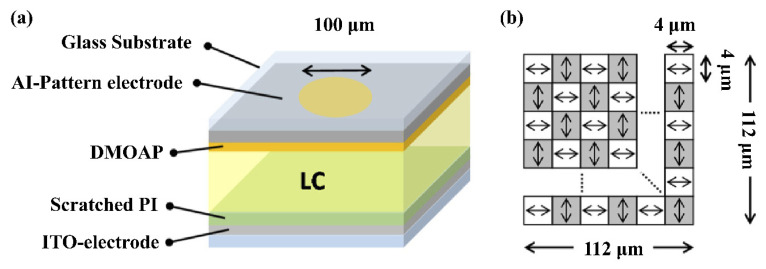
Polarizer-free LC micro-lens. (**a**) Schematic configuration (aperture of a micro-lens: 100 μm). (**b**) The illustration of a scratched PI area in (**a**) [53]. Reprinted with permission from Ref. [53]. 2021 The Optical Society.

**Figure 8 molecules-27-08025-f008:**
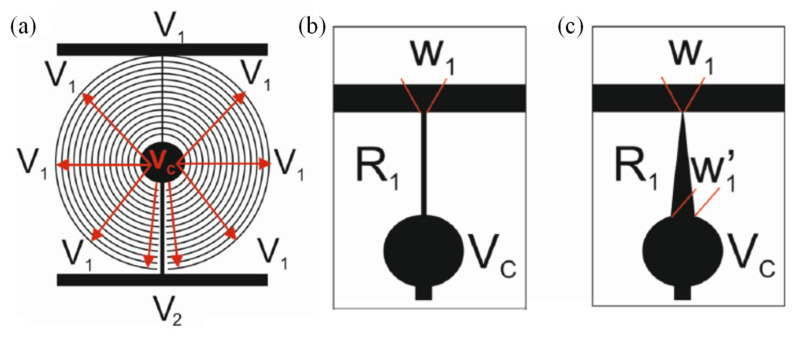
Schematic depiction of the (**a**) voltage-dividing electrode. (**b**) Rectangular transmission line. (**c**) Triangular transmission line [56]. Reprinted with permission from Ref. [56]. 2020 Springer Nature.

**Figure 9 molecules-27-08025-f009:**
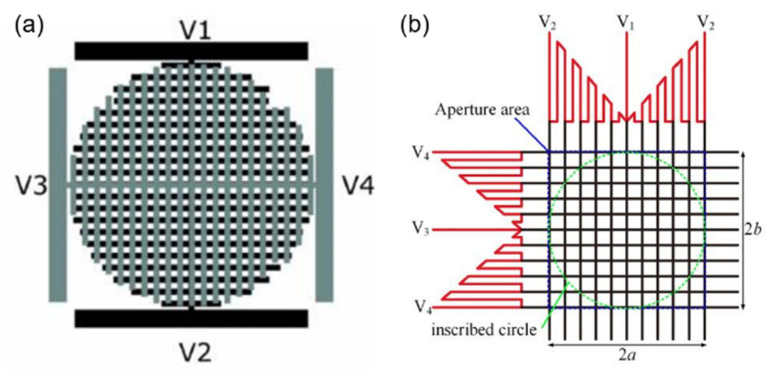
Pattern electrodes with parallel sub-electrodes. (**a**) Two electrode structures perpendicularly arranged, corresponding to the top and bottom substrates of a LC-cell [57]. (**b**) Resistance line distribution with a linear function of the coefficient *k* [63]. Reprinted with permission from Refs. [57,63]. 2020 Springer Nature and 2022 IEEE.

**Figure 10 molecules-27-08025-f010:**
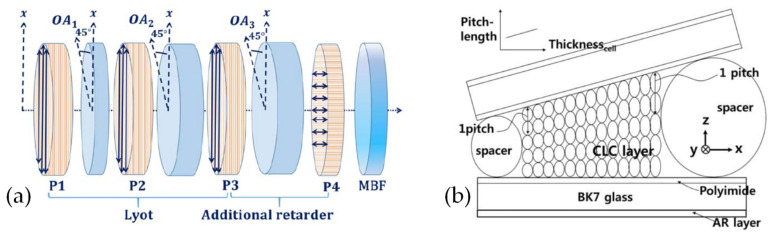
Schematic diagram of (**a**) LC filter based on a Lyot filter [69] and (**b**) the wedge CLC cell with a pitch gradient [70]. Reprinted with permission from Refs. [69,70]. 2016 The Optical Society and 2019 IEEE.

**Figure 11 molecules-27-08025-f011:**
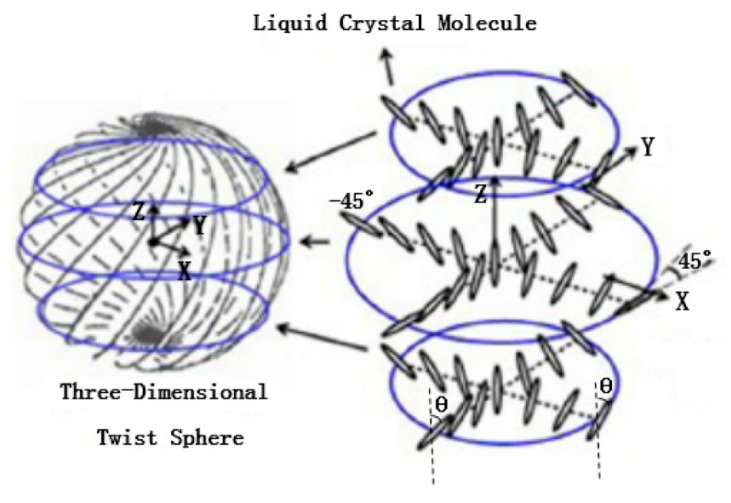
The model of the three-dimensional twist sphere structure in the sphere phase [71]. Reprinted with permission from Ref. [71]. 2019 MDPI.

**Figure 12 molecules-27-08025-f012:**
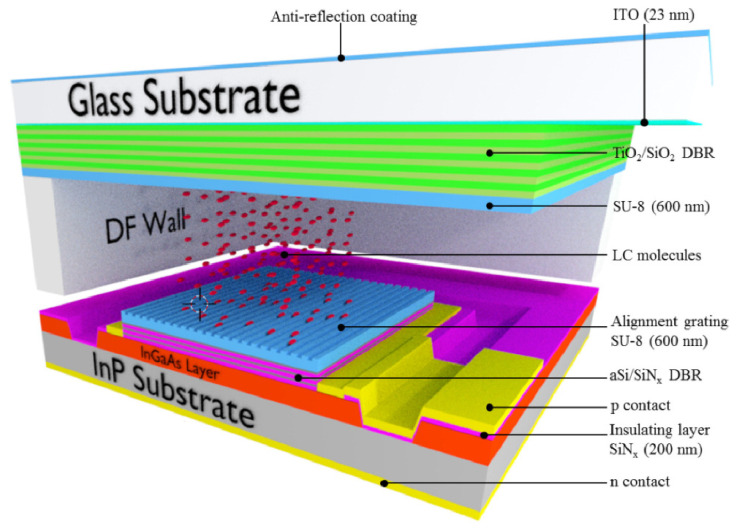
Three-dimensional and cross-section view for a single tunable PD [72]. Reprinted with permission from Ref. [72]. 2018 The Optical Society.

**Figure 13 molecules-27-08025-f013:**
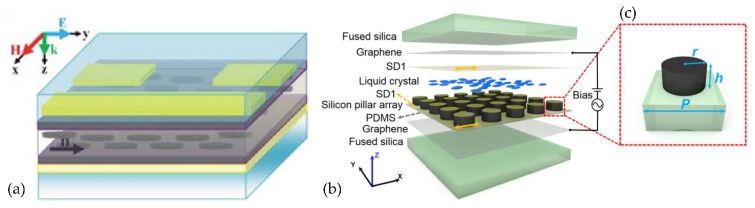
Metamaterial absorber in the THz range: (**a**) Planar Square Titanium SRR [74]. (**b**) The decomposition diagram of the LC-integrated dielectric metamaterial [76]. (**c**) The unit dimension of the pillar resonator: lattice periodicity, p: 210 μm; radius, r: 64 μm; height, h: 60 μm. Reprinted with permission from Refs. [74,76]. 2019 Taylor & Francis and 2018 MDPI.

**Figure 14 molecules-27-08025-f014:**
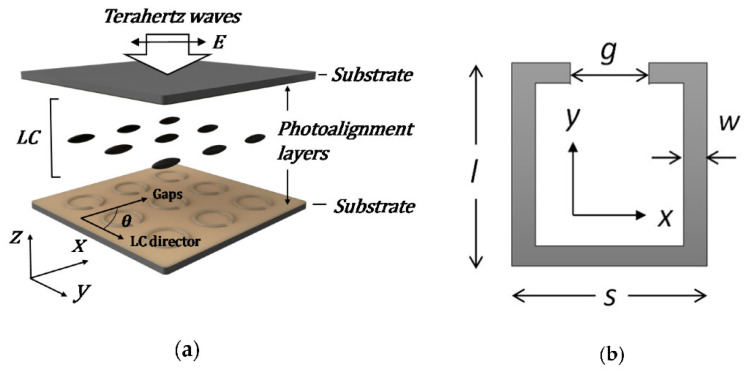
Dimensions of SRRs with different units: (**a**) split ring [73] and (**b**) split square [79]. Reprinted with permission from Refs. [73,79]. 2021 MDPI.

**Figure 15 molecules-27-08025-f015:**
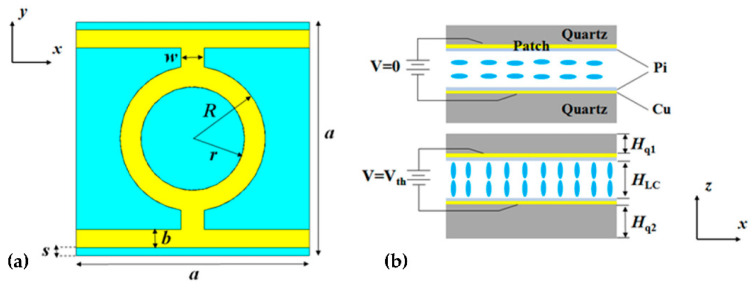
Schematic diagram of a metamaterial absorber based on liquid crystals [80]: (**a**) top view and (**b**) side view. Reprinted with permission from Ref. [80]. 2018 MDPI.

**Figure 16 molecules-27-08025-f016:**
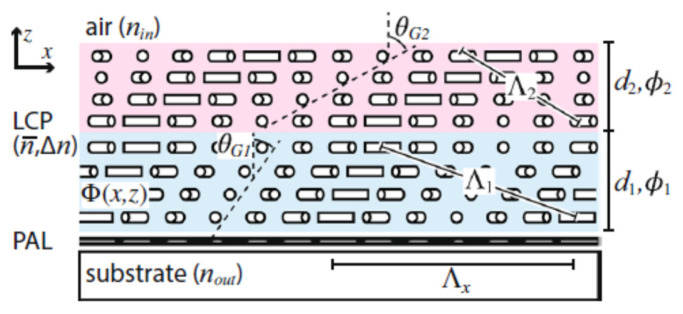
Double-layer twisted Bragg grating [84]. Reprinted with permission from Ref. [84]. 2018 Spring Nature.

**Figure 17 molecules-27-08025-f017:**
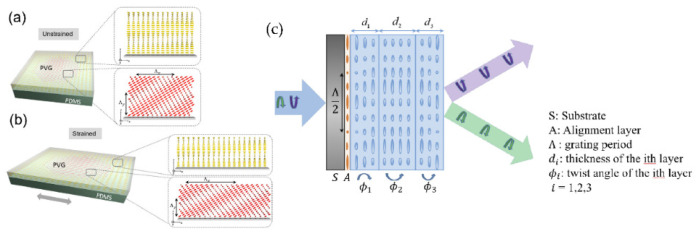
Schematic illustrations of the PVG structure and stretching process of PVG from an unstrained state to strained state. (**a**) PVG on PDMS without strain. (**b**) Strained PVG structure [90]. (**c**) Three-layer twisted structure PB phase control beam deflector [85]. Reprinted with permission from Refs. [90,85]. 2019 The Optical Society.

**Figure 18 molecules-27-08025-f018:**
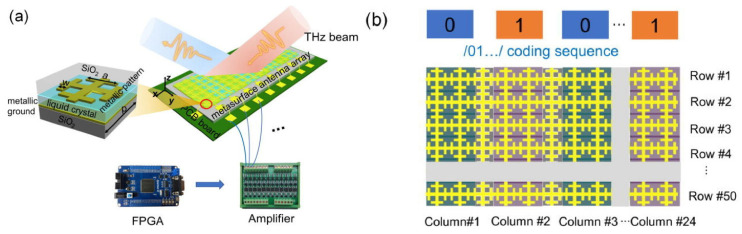
Cross-type programmable element grating. (**a**) Brief circuit schematic diagram. (**b**) Specific unit allocation control [29].

**Figure 19 molecules-27-08025-f019:**
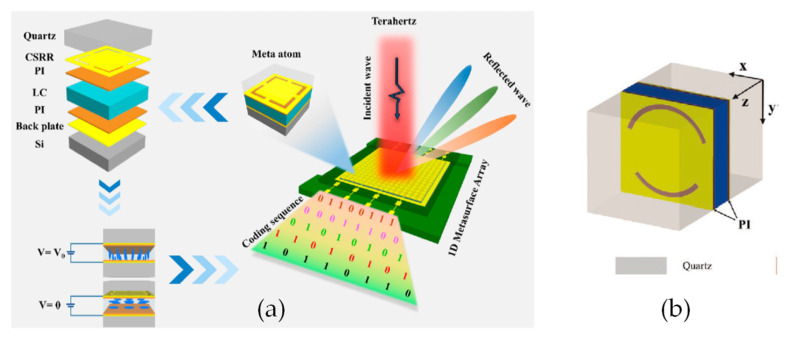
The elements of the array-grating: (**a**) square SRR [86] and (**b**) arc SRR [89]. Reprinted with permission from Refs. [86,89]. 2022 American Chemical Society and 2021 John Wiley and Sons.

**Figure 20 molecules-27-08025-f020:**
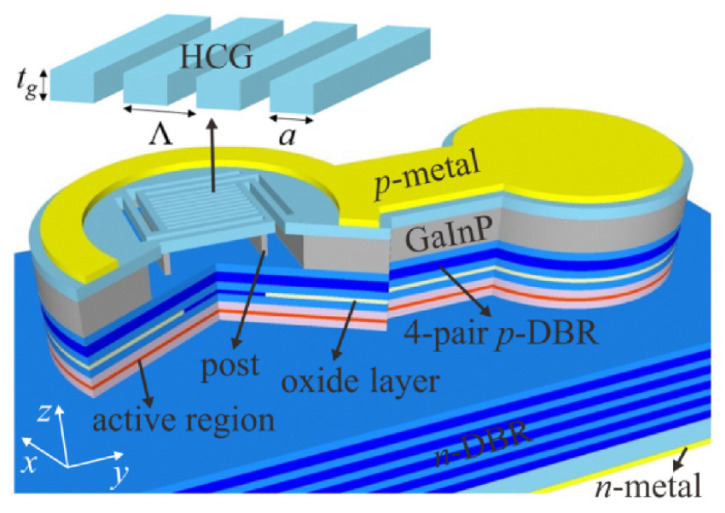
Schematic diagram of the structure. Basic structure of the tunable LC-VSCEL the same as the HCG liquid crystal laser control structure are displayed [101]. Reprinted with permission from Ref. [101]. 2022 The Optical Society.

**Figure 21 molecules-27-08025-f021:**
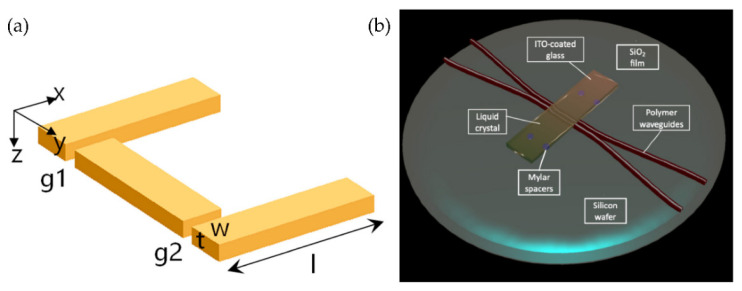
Optical switch structure: (**a**) nanometal structure [92] and (**b**) MMI channel switching control [93]. Reprinted with permission from Refs. [92,93]. 2017 Elsevier and 2019 Beilstein-Institut.

**Figure 22 molecules-27-08025-f022:**
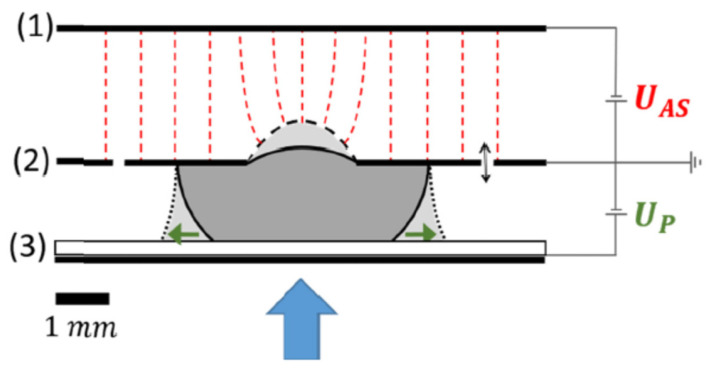
Joint control lens of electrowetting and Maxwell force. Schematic of the device consisting of three electrodes (1), (2), (3) with different voltages applied. [123]. Reprinted with permission from Ref. [123]. 2019 The Optical Society.

**Figure 23 molecules-27-08025-f023:**
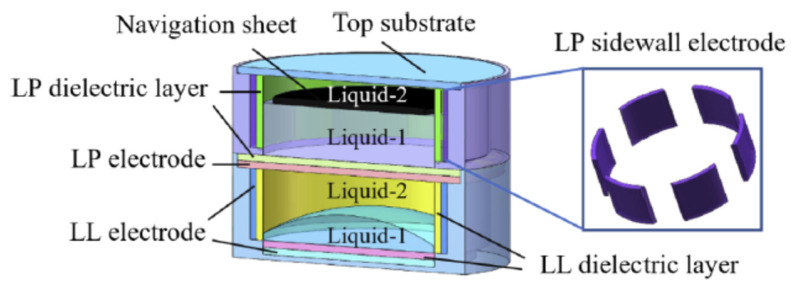
Multifunctional lens with single- and multi-electrode controls [127]. Reprinted with permission from Ref. [127]. 2020 The Optical Society.

**Figure 24 molecules-27-08025-f024:**
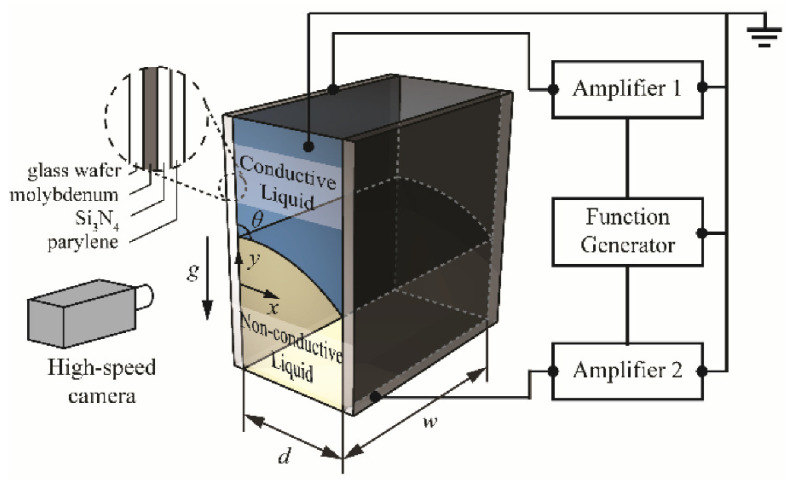
Schematics of the electrical setup to actuate the liquid–liquid interface by EWOD [128]. Reprinted with permission from Ref. [128]. 2019 Spring Nature.

**Figure 25 molecules-27-08025-f025:**
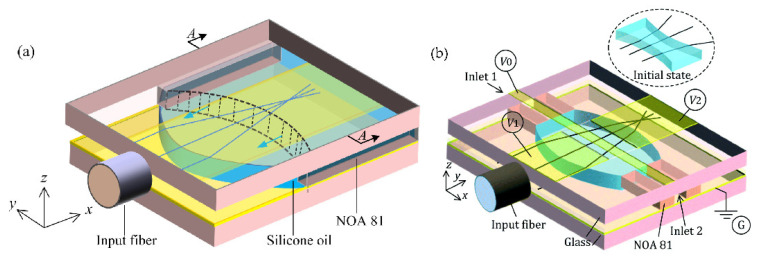
Schematic design of the DEP lens, the DEP force drives the liquid–air interface from concave to convex through: (**a**) one side [129] and (**b**) both sides [130]. Reprinted with permission from Refs. [129,130]. 2018 The Optical Society and 2018 Royal Society of Chemistry.

**Figure 26 molecules-27-08025-f026:**
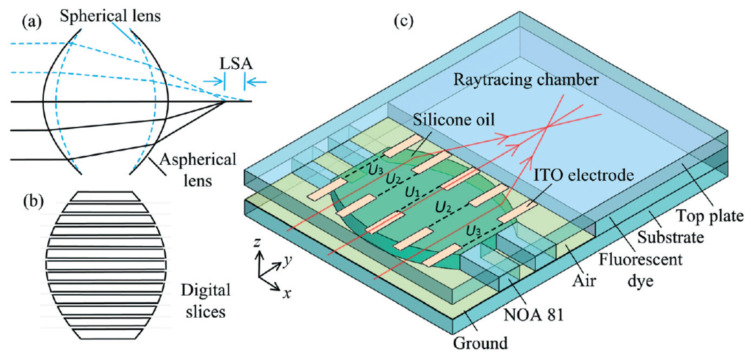
Schematic design of the DEP-actuated aspherical lens with two arrays of ITO electrode strips patterned on the top plate. (**a**) Working principle of the control of longitudinal spherical aberration (LSA). (**b**) The lens interfaces are divided into discrete slices with variable local curvatures to form an aspherical lens. (**c**) Schematic design of the DEP-actuated aspherical lens [131]. Reprinted with permission from Ref. [131]. 2020 Royal Society of Chemistry.

**Figure 27 molecules-27-08025-f027:**
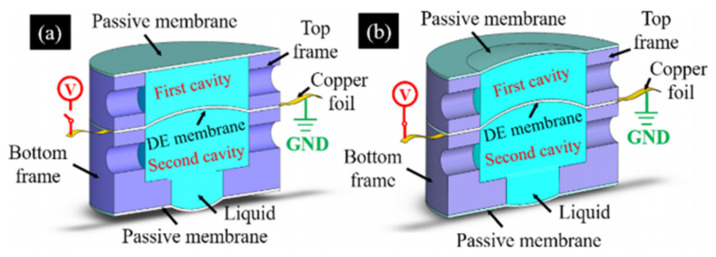
Dual-chamber sandwich DE lens: (**a**) In the initial state, the three membranes deform due to the hydraulic pressure. (**b**) Three membranes reshaped when a driving voltage is applied to the copper foil [116]. Reprinted with permission from Ref. [116]. 2020 The Optical Society.

**Figure 28 molecules-27-08025-f028:**
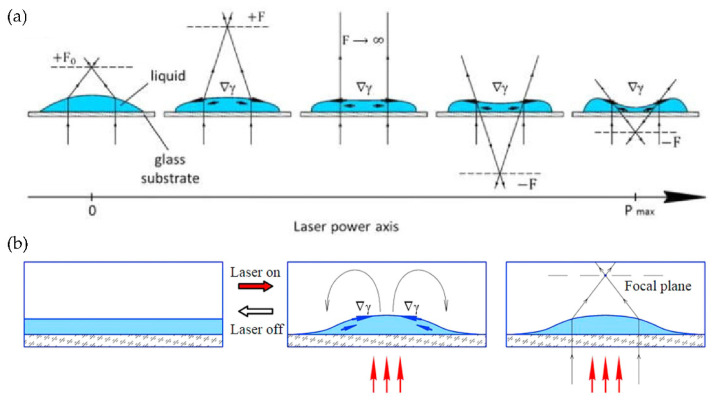
Thermal effect lens actuated by laser-induced: (**a**) Marangoni forces [133], the laser power axis shows the power increase; (**b**) solutocapillary forces [134], schema of the droplet formation process by laser, which contributes to the surface tension gradient. Reprinted with permission from Refs. [133,134]. 2018 AIP Pulishing and 2017 Elsevier.

**Figure 29 molecules-27-08025-f029:**
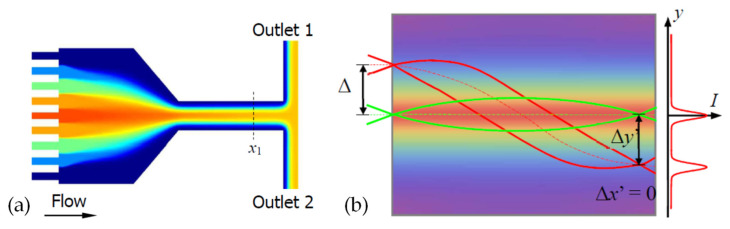
Multichannel mixing GRIN lens. (**a**) The structure to form the refractive index distribution, and (**b**) the equivalent optical path diagram of the graded refractive index lens [135]. Reprinted with permission from Ref. [135]. 2016 Royal Society of Chemistry.

**Figure 30 molecules-27-08025-f030:**
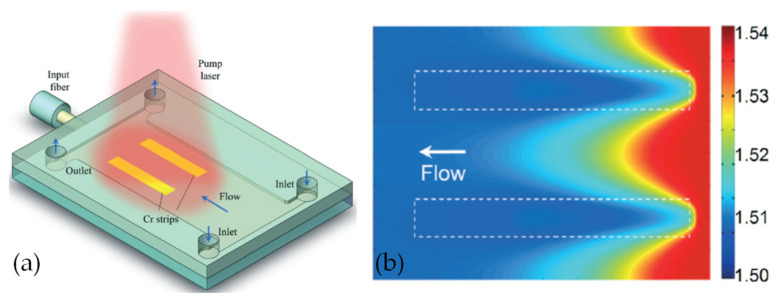
Laser-induced thermal gradient. (**a**) Schematic diagram of the thermal lens, and (**b**) the simulated two-dimensional refractive index profiles [136]. Reprinted with permission from Ref. [136]. 2016 Royal Society of Chemistry.

**Figure 31 molecules-27-08025-f031:**
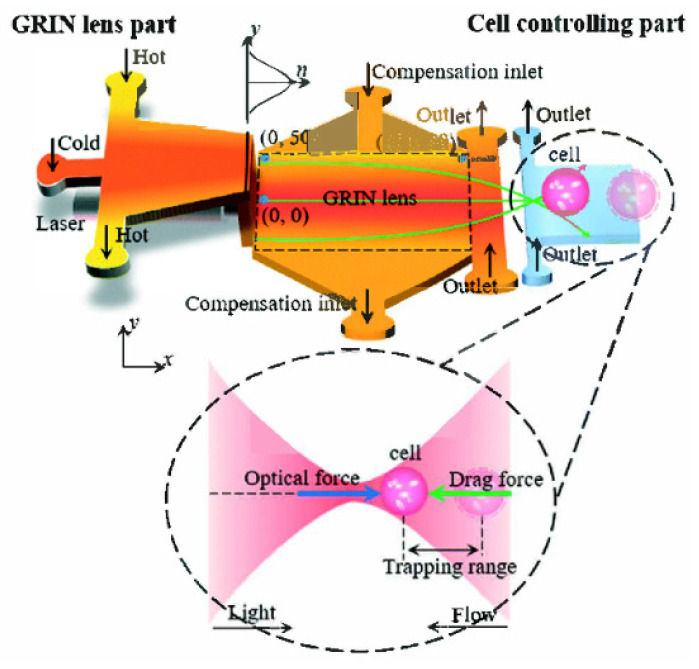
The schematic design of fluidic thermal GRIN lens for cell manipulation: The system includes a lens chamber and a cell trapping chamber. Five streams at different temperatures are injected into the microfluidic chip to form a gradient refractive index across the channel [137]. Reprinted with permission from Ref. [137]. 2017 Royal Society of Chemistry.

**Figure 32 molecules-27-08025-f032:**
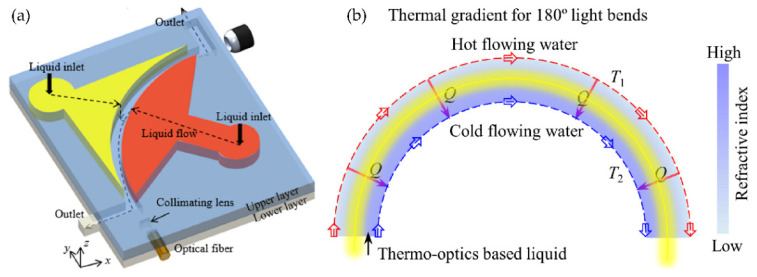
Diffusion device: (**a**) Counterflow convection–diffusion, Refractive index profile in a concentration gradient light-bending device [43]. (**b**) thermal-gradient bending device, conceptual design of the concentration gradient bending device and refractive index profile in a thermal gradient light-bending device [44]. Reprinted with permission from Refs. [43,44]. 2017 and 2020 The Optical Society.

**Figure 33 molecules-27-08025-f033:**
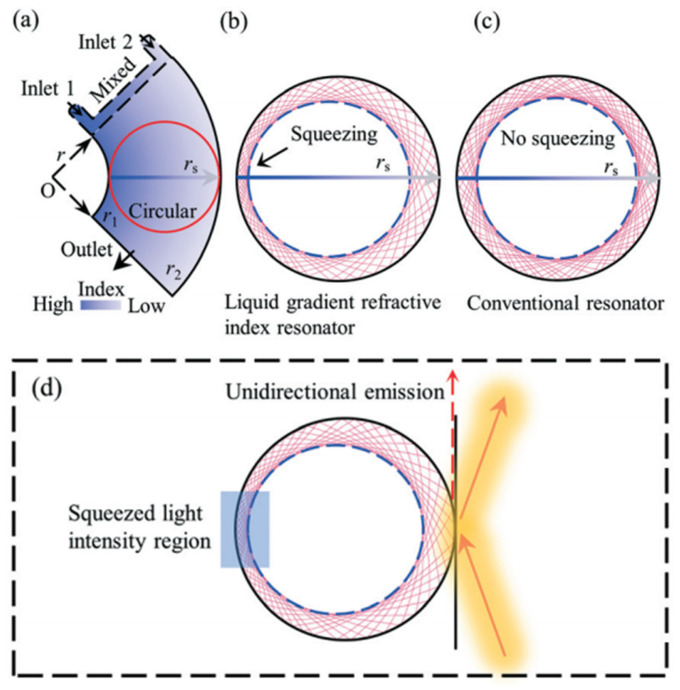
Schematic of a liquid–diffusion optofluidic gradient index resonator. (**a**) The optofluidic diffusing structure, (**b**) ray trajectory squeezed and (**c**) ray trajectories are symmetrical in a conventional circular cavity. (**d**) Squeezing the light intensity profile enables unidirectional emissions on the low refractive index side [138]. Reprinted with permission from Ref. [138]. 2020 Royal Society of Chemistry.

**Figure 34 molecules-27-08025-f034:**
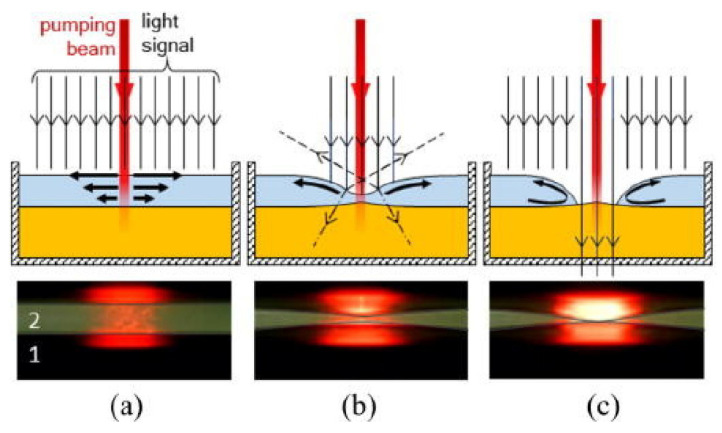
Schematic diagram of the working mechanism. (**a**) The thermocapillary flow rises in the upper layer. (**b**) The thermocapillary deformation of the upper layer and the convex deformation of the bottom layer. (**c**) The upper thermocapillary rupture [139]. Reprinted with permission from Ref. [139]. 2019 AIP Publishing.

**Figure 35 molecules-27-08025-f035:**
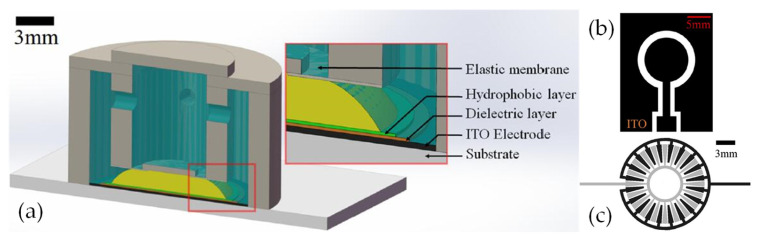
Phase modulator structure. (**a**) Cross-sectional view of the liquid phase modulator; the bottom electrode configuration: (**b**) key type; (**c**) interdigital type. Reprinted/adapted with permission from Ref. [142]. 2021 American Chemical Society.

**Table 1 molecules-27-08025-t001:** A summarization of the LC device with typical references attached.

Device Type	Classification	Working Principle	Materials
Adaptive LC lens	Modal type	Electronically	NLC [47]
	Pattern type		NLC [56]
Tunable LC filter/absorbers	Visible to infrared	Electronically	SPLC [71], CLC [70], NLC [68]
	THz		NLC [73,80], PDLC [77]
LC Beam controllers	Deflector	Electronically	CLC [84], NLC [29,89]
	Laser polarization	Light-operated	NLC [100]
	Optical switch	Electronically	NLC [93]
Bistable state LC devices	E-paper	Electronically	PDLC [103], TNLC [106], CLC [107]
	Smart windows	Electronically, light-operated	PDLC [114], CLC [109], PSLC [108]

**Table 2 molecules-27-08025-t002:** A summarization of an isotropic liquid device with typical references attached.

Device Type	Working Principle	Device Type	Materials
Interface modulation	EW	Tunable lens, deflector	Silicone oil [126]
	DEP	Tunable lens, e-paper	Silicone oil [129,130]
	DE membranes	Tunable lens	Glycerol [118]
	Thermal	Tunable lens, beam tracker	Ethanol and glycol [133,134]
Index distribution	Diffusion	GRIN lens	Benzyl alcohol [136]
		GRIN resonator	Benzyl alcohol and glycol [138]
		Bending waveguide	Glycol [43]

## Data Availability

Not applicable.

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
