# Peer review of "Recent Development of Tunable Optical Devices Based on Liquid"

_molecules, 2022, doi:10.3390/molecules27228025_

Round 1

Reviewer 1 Report

This manuscript reviewed the recent development of tunable optical devices based on isotropic liquids and anisotropic liquid crystals. The topic is attractive and the references are up to date. The manuscript is suitable for publication in this journal after addressing the following issues:

1.    It’s recommended to add a figure in the introduction part to summarize the principles of liquid optics control and the types of the optical devices based on the corresponding principle.

2.    There are some tunable optical devices such as electronic paper and smart windows, which are also based on liquid or liquid crystal materials. It’s better to make a introduce of them in the main text to make the review more comprehensive.

3.    It’s recommended to supplement a table to summarize the different types of optical devices, their working principles, the materials used for preparation of the devices and the related references.

Reviewer 2 Report

In this review-type manuscript, the authors review the tunable liquid devices, including isotropic liquid and anisotropic liquid crystal devices. And the limitations and future perspectives of current liquid device are discussed. For me, this is a nice review paper and I recommend its acceptance to Molecules. Before publication, the following minor points should be corrected.

1.     The authors start this review paper with the discussion of tunability of optical properties but without citing any publications on the first paragraph of the Introduction section. Some necessary papers should be cited. Like, the research expansion of wavelength in the optical field is a disastrous challenge to solid optics. And nonlinear optical crystals are the one of the key materials to expand the wavelength, related review papers, like 10.1002/anie.201913974 and 10.1021/acs.chemrev.0c00796 are suggested to cite.

2.     In the section of Conclusion and Future. The authors list some existing problems, is that possible to give some discussion about the solution of these urgent issues related to tunable liquid device.
